

# Evaluating and improving modeled turbulent heat fluxes across the North American Great Lakes

Umarporn Charusombat[1], Ayumi Fujisaki-Manome[2,3], Andrew D. Gronewold[1], Brent M.
Lofgren[1], Eric J. Anderson[1], Peter D. Blanken[4], Christopher Spence[5], John D. Lenters[6],
Chuliang Xiao[2], Lindsay E. Fitzpatrick[2], and Gregory Cutrell[7]

[1]NOAA Great Lakes Environmental Research Laboratory, Ann Arbor, Michigan, 48108
USA
[2]University of Michigan, Cooperative Institute for Great Lakes Research, Ann Arbor,
Michigan, 48108, USA
[3]University of Michigan, Climate & Space Sciences and Engineering Department, Ann
Arbor, Michigan, 48109, USA
[4]University of Colorado, Department of Geography, Boulder, Colorado, 80309, USA
[5]Environment and Climate Change Canada, Saskatoon, Saskatchewan, S7N 5C5, Canada
[6]University of Wisconsin-Madison, Center for Limnology, Boulder Junction, Wisconsin,
54512, USA
[7]LimnoTech, Ann Arbor, Michigan, 48108

*Correspondence to: Ayumi Fujisaki-Manome (ayumif@umich.edu)*





**Abstract** Turbulent fluxes of latent and sensible heat are important physical processes that influence the energy and water budgets of the North American Great Lakes. Validation and improvement of bulk flux algorithms to simulate these turbulent heat fluxes are critical for accurate prediction of lake hydrodynamics, water levels, weather, and climate over the region. Here we

5     consider five heat flux algorithms from three parent model systems; the Finite-Volume Community Ocean Model (FVCOM, with three different options for heat flux algorithm), the Weather Research and Forecasting (WRF) model, and the Large Lake Thermodynamics Model, which are used in research and operational environments and concentrate on different aspects of the Great Lakes' physical system. The heat flux algorithms were isolated from each model and

10     driven by meteorological data from four over-lake stations within the Great Lakes Evaporation Network (GLEN). The simulation results were then compared with eddy covariance flux measurements from the same GLEN sites. All algorithms reasonably reproduced the seasonal cycle of the turbulent heat fluxes while the original algorithms except for the Coupled Ocean Atmosphere Response Experiment (COARE) algorithm showed notable overestimation of the

15     fluxes in fall and winter. Overall, COARE had the best agreement with eddy covariance measurements. Simulations with the four algorithms other than COARE were improved by updating the parameterization of roughness length scales for air temperature and humidity to match those used in COARE. Agreement between modeled and observed fluxes notably varied according to the geographic locations of the GLEN sites.





## 1. Introduction

Simulating physical processes within and across large freshwater water bodies is typically achieved using oceanographic-scale models representing heat and mass exchange below, above, and across the air-water interface. Verification and skill assessment of these models is limited, however, by the quality and spatial extent of observations and data. The datasets available for validation of ocean dynamical models, for example, include satellite-based surface water temperatures (Reynolds et al., 2007), sea surface height (Lambin et al., 2010), and, when available, *in situ* measurements of sensible and latent heat fluxes (Edson et al., 1998). Dynamical and thermodynamic models for large lakes are often verified using similar measurements (Chu et al., 2011; Croley, 1989a, 1989b; Moukomla and Blanken, 2017; Xiao et al., 2016; Xue et al., 2016). However the spatiotemporal resolution of *in situ* measurements for these variables in lakes is comparatively sparse (Gronewold and Stow, 2014), particularly for latent and sensible heat fluxes.

On the Laurentian Great Lakes (hereafter referred to as the Great Lakes), sensible and latent heat fluxes play an important role in the seasonal and interannual variability of critical physical processes including spring and fall lake evaporation (Spence et al., 2013), the onset, retreat, and spatial extent of winter ice cover (Van Cleave et al., 2014; Clites et al., 2014), and air-mass modification including processes such as lake-effect snow (Wright et al., 2013). These phenomena, in turn, impact lake water levels (Gronewold et al., 2013; Lenters, 2001), both atmospheric and lake circulation patterns (Beletsky et al., 2006), and the fate and transport of watershed-borne pollutants (Michalak et al., 2013). For decades, Great Lakes dynamical and thermodynamic models simulating these processes have done so without the benefit of observations.

The Finite-Volume Community Ocean Model (FVCOM), for example, is a widely used hydrodynamic ocean model that that has been found to provide accurate real-time nowcasts and forecasts of hydrodynamic conditions across the Great Lakes including 3D currents, water temperature, and water level fluctuations across relatively fine spatiotemporal scales (Anderson et al., 2015; Anderson and Schwab, 2013; Bai et al., 2013; Xue et al., 2016). FVCOM is currently being developed, tested, and deployed across all of the Great Lakes as part of an ongoing update to the National Oceanic and Atmospheric Administration (NOAA) Great Lakes Operational Forecasting System (GLOFS). To date, however, there has been no direct verification of the turbulent heat flux algorithms intrinsic to FVCOM; this is an important step, in light of the fact that FVCOM flux algorithms were developed primarily for the open ocean and, until now, have



been assumed to provide reasonable turbulent heat flux simulations across broad freshwater surfaces as well.

The Large Lake Thermodynamic Model (Croley, 1989a, 1989b; Croley et al., 2002; Hunter et al., 2015), LLTM, is a conventional lumped conceptual lake model. It is employed in seasonal operational water supply and water level forecasting by water resource and hydropower management authorities (Gronewold et al., 2011) and is used as a basis for long-term historical monthly average evaporation records (Hunter et al., 2015). It has historically been calibrated and verified using observed ice cover and surface water temperature, but not turbulent flux, data. Among more complex atmosphere-lake model systems, the Weather Research and Forecasting (WRF) system is increasingly used in Great Lakes applications (Xiao et al., 2016; Xue et al., 2015). However, only with the Global Environmental Multiscale model (GEM, (Bélair et al., 2003a, 2003b; Deacu et al., 2012)), observed turbulent fluxes have been employed to assess predictive skill of net basin supply and regional climate conditions (Deacu et al., 2012).

To address this gap in the development and testing of experimental and operational Great Lakes physical models, we employ data from a network of relatively novel year-round offshore flux measurements collected over the past decade at lighthouse-based eddy covariance towers. Specific foci in this study are to determine: 1) the capability of the flux algorithms in reproducing inter-annual, seasonal, and daily latent and sensible heat fluxes, 2) how much variability occurs in the simulated latent and sensible heat fluxes from using different flux algorithms with common forcing data (e.g. meteorology and water surface temperature), and 3) the source of such variability and simulation errors. In particular, we will address how different parameterizations of roughness length scales affect simulations of turbulent latent and sensible heat fluxes over the water surface of the Great Lakes.

## 2. Methods

We begin by describing observational data of meteorology and turbulent heat fluxes used in this study, the flux algorithms within larger modelling frameworks, and intercomparison methods used to evaluate the performance of the flux algorithms. We selected the time period from January 2012 through December 2014; this period is ideally suited for our study since it allows for a



comparison between two unusually warm (2012-2013) and unusually cold (2013-2014) winters (Clites et al., 2014).

## 2.1. Data

Meteorological and turbulent heat flux data for this study were collected from four offshore, lighthouse-based monitoring platforms (Fig. 1): Stannard Rock (Lake Superior), White Shoal (Lake Michigan), Spectacle Reef (Lake Huron), and Long Point (Lake Erie).  These observations are collected as part of a broader collection of fixed and mobile-based platforms collectively referred to as the Great Lakes Evaporation Network (GLEN, Lenters et al., 2013). These

installations are co-located with NOAA National Data Buoy Center (NDBC) stations STDM4, WSLM4, and SRLM4 at Stannard Rock, White Shoal, and Spectacle Reef, respectively.

     With the exception of Long Point, footprint analysis indicates each station is located sufficiently distant from shore that terrestrial-aquatic boundary conditions do not influence our measurements. (Flux footprint calculations show this to be predominantly the case during most times of the year,

Blanken et al., 2011). Long Point, however, is located at the tip of a narrow, 40-km peninsula extending into Lake Erie.  As a result, measured fluxes can be influenced by the upwind land surface when the wind direction is between 180$^{\circ}$ and 315$^{\circ}$), so these data were removed when measured wind directions were within this range.

2.1.1. *Turbulent heat flux measurements*

     All four eddy covariance systems follow conventional protocols for calculating turbulent fluxes, such as those established on Great Slave Lake (Northwest Territories, Canada) by Blanken et al. (2000). 30-minute mean turbulent fluxes of sensible and latent heat ($\lambda E$ and $H$, respectively; W m$^{-2}$; positive upward from the surface) were calculated from 10-Hz measurements of the

vertical wind speed ($w$; m/s), air temperature ($T$; $^{\circ}$C), and water vapor density ($\rho_v$; g/m$^3$). Wind speed was measured using a 3D ultrasonic anemometer (Campbell Scientific CSAT-3), while water vapor density was measured using a krypton hygrometer (Campbell Scientific KH20). The statistics (means and covariances) of the high-frequency data were collected and processed at 30-minute intervals using a Campbell Scientific dataloggers. Corrections to the eddy covariance

measurements included 2D coordinate rotation (Baldocchi et al., 1988), and corrections for air density fluctuations (Webb et al. 1980), sonic path length, high-frequency attenuation, and sensor


separation (Horst, 1997; Massman, 2000). Instrument heights for meteorological and the eddy covariance measurements were 32.5 m at Stannard Rock, 29.5 m at Long Point, 30.0 m at Spectacle Reef, and 42.8 m at White Shoal.

As noted in section 2.1, the eddy covariance data at Long Point were filtered out when wind direction was between $180^{\circ}$-$315^{\circ}$ to remove the land surface influence on the measured latent and sensible heat fluxes. We also applied cross-check filtering for the eddy covariance data at White Shoal and Spectacle Reef. The two stations were relatively close in distance and the measured latent and sensible heat fluxes at these stations were mostly similar, with daily averaged values differing by less than 100 W m$^{-2}$ (except during the ice-covered periods, which were not a focus

of this study). There were outliers during July and August 2014 where the measured fluxes differed at the two stations by greater than 100 W m$^{-2}$. These data were removed, resulting in ~5% loss of data points at White Shoal and Spectacle Reef. See Blanken et al. (2011) and Spence et al. (2011, 2013) for details of the measurements and flux corrections.

### 2.1.2. Meteorological data and water surface temperature

     At the same heights as the turbulent flux instruments, half-hourly meteorological variables of wind speed, air temperature, relative humidity, and air pressure were obtained using RM Young wind sensors, Vaisala HMP45C thermohygrometers, and the CSAT-3. Air pressure data at the

Spectacle Reef station were missing from the record and were approximated using data from the White Shoal station. Water surface temperature for model input is taken from Great Lakes Surface Environmental Analysis (GLSEA, https://coastwatch.glerl.noaa.gov/glsea/doc/), which is a composite analysis based on NOAA Advanced Very High Resolution Radiometer (AVHRR) imagery. Lake surface temperatures are updated daily with an interpolation method using

information from the cloud-free portions of the satellite imagery within ±10 days. The closest pixels to the observation sites were chosen to provide model inputs of water surface temperature. Ice concentration data provided by the National Ice Center (NIC) were used to decide whether eddy covariance measurements at each GLEN site were affected by ice cover. When ice concentration at the closest pixel to a GLEN station was greater than zero, we did not use any data

for our comparison (i.e. the observed heat fluxes, water surface temperature, and meteorological





data). This was because the study focused on evaluating the turbulent heat fluxes over water during ice-free periods.

Infrared thermometers (IRTs, Apogee IRR-T) were also installed on the observation platforms to measure water surface temperature. However, test simulations showed that the flux values

simulated using the water surface temperature from the IRTs were generally less reliable than when using the GLSEA data. Blanken et al. (2011) found that about 30% of the IRT-measured lake surface temperature observations were unreliable due to condensation, frost, and interference from other surfaces (e.g., the lighthouse or sky). It is likely that this issue affected the accuracy of IRT-measured water surface temperature during the period of our study. Therefore, we did not use

the IRT-based measurements of water surface temperature as input to the simulations.

### 2.2. Flux algorithms

Five different flux algorithms were evaluated in this study. These algorithms are incorporated into hydrodynamic/atmospheric/hydrologic models that are frequently used for Great Lakes

operational and research applications (Fig. 2).

In an early stage of its development, FVCOM required prescribed heat fluxes as forcing (Chen et al., 2006a, 2006b). The Coupled Ocean Atmosphere Response Experiment (COARE) Met Flux Algorithm version 2.6 (Fairall et al., 1996a,b) was first adopted in the official FVCOM version 2.7 (Chen et al., 2006a). The COARE Met Flux Algorithm is one of the most frequently used

algorithms in the air–sea interaction community. It was subsequently modified and validated at higher winds in the version known as COARE 3.0 (Fairall et al., 2003) and the latest version COARE 3.5 (Edson et al., 2013) includes wave influences on the Charnock parameter (Charnock, 1955). FVCOM mostly incorporated these updates as the model was upgraded, including provision for freshwater implementation, except that the latest version of FVCOM (version 4.0) has not yet

included wave influences on the Charnock parameter. Hereafter we refer to the COARE implementation in FVCOM as COARE. In FVCOM version 3 and later, two additional flux calculation algorithms were added (Chen et al., 2013): One was adapted from a flux coupler in the Community Earth System Model (CESM, Jordan et al. 1999; Kauffman and Large 2002) and also built into the code of the Los Alamos Sea Ice Model (CICE, Hunke et al., 2015). This algorithm

will hereafter be referred to as J99 hereafter (i.e., Jordan et al., 1999). The other algorithm, hereafter referred to as LS87 (Liu and Schwab, 1987) was originally developed at NOAA's Great



Lakes Environmental Research Laboratory (GLERL) and subsequently used in a variety of Great Lakes research and operational applications (Beletsky et al. 2003; Wang et al. 2010; Anderson and Schwab 2013; Rowe et al. 2015; and many others). Inclusion of LS87 in FVCOM was tied to the fact that the algorithm was historically part of real-time nowcasts and forecasts of NOAA's

GLOFS, which is based on the Princeton Ocean Model, and that GLOFS is transitioning its physical model to FVCOM.

The WRF model (Skamarock et al., 2008) is increasingly used for regional weather and climate model applications over the Great Lakes (Benjamin et al., 2016; Xiao et al., 2016; Xue et al., 2015). The WRF model includes a one-dimensional lake model that thermodynamically interacts

with the overlaying atmosphere (WRF-lake, Bonan, 1995; Gu et al., 2015; Henderson-Sellers, 1986; Hostetler and Bartlein, 1990; Hostetler et al., 1993; Subin et al., 2012) and is adapted from the Community Land Model version 4.5 (Oleson et al., 2013; Zeng et al., 1998). The algorithm for the turbulent heat flux calculation in WRF-lake is mainly based on Zeng et al., (1998) with a difference in roughness scale parameterization for its WRF-lake application. Hereafter, this

algorithm is referred to as Z98L).

Finally, we include the flux algorithm from the LLTM (Croley, 1989a,b; Croley et al., 2002; Hunter et al., 2015), which is a lumped conceptual lake model that was developed for hydrological research and forecasting for the Great Lakes. LLTM is developed to simulate evaporation and heat fluxes as a lake-wide average, rather than spatially distributed. This algorithm is based primarily

on the work of Croley et al. (1989a,b) and is hereafter referred to as C89.

All of the above algorithms are based on applications of Monin-Obukhov similarity theory (Kantha and Clayson, 2000b; Obukhov, 1971), where the turbulent fluxes of sensible heat, latent heat, and momentum are expressed with scaling parameters $\theta^*$, $q^*$, $u^*$ for potential air temperature, specific humidity, and horizontal wind velocity, respectively. In each algorithm, the major

differences are in the derivation of the bulk transfer coefficients, $C_H$ and $C_E$ for the sensible heat and the latent heat, respectively in the bulk expressions to calculate the sensible and latent heat fluxes:

$$H = \rho_a c_p C_H S(\theta_w - \theta_a), \qquad (1)$$

$$\lambda E = \rho_a \lambda C_E S(q_w - q_a), \qquad (2)$$





where $\rho_a$ is the density of air; $c_p$ and $\lambda$ are the specific heat of air and the latent heat of vaporization, respectively; $S$ is the average value of wind speed (defined later); and $\theta_w$ and $\theta_a$ ($q_w$ and $q_a$) are potential temperature (specific humidity) of the water surface and of air at the measurement height, respectively.

The transfer coefficients have a dependence on atmospheric stability that can be expressed as:

$$C_D = \kappa \left[ \ln \left( \frac{z}{z_0} \right) - \Psi_M(\zeta) \right]^{-2} \tag{3}$$

$$C_{H,E} = \kappa Pr_t^{-1} \left[ \ln \left( \frac{z}{z_0} \right) - \Psi_M(\zeta) \right]^{-1} \left[ \ln \left( \frac{z}{z_{0\theta,0q}} \right) - \Psi_{\theta,q}(\zeta) \right]^{-1} \tag{4}$$

$$u^* = S \sqrt{C_D} \tag{5}$$

$$\theta^* = \Delta\theta \sqrt{C_H} \tag{6}$$

$$q^* = \Delta q \sqrt{C_E}, \tag{7}$$

where $z_0$, $z_{0\theta}$, and $z_{0q}$ are roughness length scales for momentum, temperature, and humidity
respectively; $C_D$ is the drag coefficient; $\kappa$ is von Kármán constant; $Pr_t$ is the turbulent Prandtl number; and $\Psi_{M,\theta,q}(\zeta)$ are the integrated forms of stability functions for momentum, temperature, and humidity, respectively. All algorithms assume that temperature and humidity have a common value of $\Psi$, i.e. $\Psi_\theta = \Psi_q = \Psi_M$ $\zeta = z/L$ is the stability factor, where $L$ is the Obukhov length and $z$ is the measurement height.

Differences among the algorithms are primarily in how they estimate $\Psi_{M,\theta,q}(\zeta)$ and $z_0$. The profile functions $\Psi_{M,\theta,q}(\zeta)$ are typically divided into three regimes, namely unstable, mildly stable, and strongly stable. All the algorithms use Businger-type parameterizations (Businger et al., 1971; Kraus and Businger, 1995) for the unstable regime (Table 1), except COARE which includes convective behaviour in highly unstable conditions by introducing a stability function for a
convective limit (Fairall et al., 1996a; Supplemental Tables 1 and 2). For stable conditions, Holtslag et al. (1990) is used in LS87, C89, and Z98L while Beljaars and Holtslag (1991) is used in J99 and COARE (Table 1).



Note that there are minor differences in coefficients of $\Psi_{M,\theta,q}(\zeta)$ in the algorithms, which can be found in Supplemental Tables 1 and 2.

The roughness length scale for momentum, $z_0$, is parameterized as a function of friction velocity $u^*$ or the average value of wind speed $S$. Note that $u^*$ is a variable diagnosed through iterative

calculations in the algorithms. The LS87, C89, and COARE algorithms apply Charnock's formula (Charnock, 1955; Smith, 1988):

$$z_0 = \frac{\alpha u^{*2}}{g} + \frac{0.11\nu}{u^*} \quad , \tag{8}$$

where $z_0$ is the roughness length scale of momentum, $\alpha$ is the Charnock parameter, $g$ is the acceleration due to gravity, and $\nu$ is kinematic viscosity. Here, COARE calculated the Charnock parameter $\alpha$ as a function of wind speed, while LS87 and C89 use a constant $\alpha$ (Table 1). J99 directly calculates $z_0$ as a function of wind speed based on Large and Pond (1981). Z98L, however, assumes $z_0$ as a constant 0.001 m. In the original paper of Zeng et al. (1998), non-constant

parameterizations for roughness length scales were used, namely Smith (1988) for momentum and Brutsaert (1982) for temperature and humidity. The constant value in Z98L is likely related to the fact that the implementation in WRF handles the lake surface as part of various land surface types, whose roughness lengths for momentum are often assumed to be constant (Mitchell et al., 2005; Oleson et al., 2013), while the original work of Zeng et al. (1998) assumed ocean surface

applications.

Evidence suggests $z_0$ can be significantly larger than $z_{0\theta,q}$, because momentum is transported across the air-sea interface by pressure forces acting on roughness elements, while heat and water vapor must ultimately be transferred by molecular diffusion across the interfacial sublayer (Brutsaert, 1975; Garratt, 1992; Kantha and Clayson, 2000a). However, many land and lake

models, including four of the five algorithms used in this study, assume the same roughness length for momentum and heat transfer; for example, Croley (1989b, C89); Liu and Schwab (1987, LS87); Oleson et al. (2013); Zeng et al. (1998, Z98L); the CICE application (J99), the previous NCEP Eta model described in Chen et al., 1997; Canadian operational weather and hydrologic models described in Deacu et al., 2012). Deacu et al. (2012) showed that the same value for $z_0$ and

$z_{0\theta,q}$ resulted in overestimation of turbulent heat fluxes over Lake Superior, and that the





overestimation was reduced by using the smooth surface parameterization for $z_{0\theta,q}$, with an empirical coefficient based on Beljaars (1994).

As part of the current study, we intend to conduct a similar experiment to Deacu et al. (2012), namely, updating the original $z_{0\theta,q}$ parameterization in the LS87, C89, Z98L, and J99 algorithms to a more realistic parameterization. We conduct this experiment to identify errors in $\lambda E$ and $H$ simulations with the original $z_{0\theta,q}$ parameterization and to evaluate how much the errors could be reduced with a more realistic parameterization. In this study, we use an alternative $z_{0\theta,q}$ formulation that is based on Fairall et al. (2003), which is used in COARE. The formulation utilizes the Liu–Katsaros–Businger model (LKB; Liu et al., 1980), with updates described in Fairall et al. (2003), where a simpler empirical relationship was formulated to represent the LKB model, based on a fit to observational data:

$$z_{0\theta,q} = \min\left(1.6{\times}10^{-4}, 5.8{\times}10^{-5} Rr^{-0.72}\right),$$ (9)

where $Rr=u^{*}z_{0}/v$ is the roughness Reynolds number. We test both of the original and updated parameterizations for $z_{0\theta,q}$ in the heat flux simulations.

"Gustiness" velocity $w_g$ is included in Z89L and COARE to account for the additional flux induced by the convective boundary layer in low wind speed regimes. The average value of wind speed $S$ is defined as,

$$S = \sqrt{U^2 + w_g{}^2},$$ (10)

where $U$ is the mean horizontal wind speed. $w_g$ is defined as

$$w_g = \beta \frac{g}{\rho_a}\left[\frac{H}{c_p T} + 0.61E\right].,$$ (11)

where $\beta$ is an empirical constant set to $\beta=1.2$ in COARE and $\beta=1.0$ in Z89L. Further details of the gustiness velocity formulations are described by Fairall et al. (1996a). In LS87, C89, and J99, $S$ is assumed to be identical to $U$.




All algorithms require meteorological inputs of horizontal wind speed $U$, potential air temperature $\theta_a$, potential temperature at the water surface temperature $\theta_w$, a humidity-related variable, air pressure, and sensor height. These meteorological inputs should represent a temporal mean field over the corresponding eddy covariance measurement. $U$, $\theta_a$, and $\theta_w$ can be directly

used in eqs. (1) and (2), while $q_w$ and $q_a$ need to be derived from relative humidity, water surface (or air) temperature, and air pressure.

### 2.3. Intercomparison methods

We take the following steps to compare and verify simulated sensible and latent heat fluxes against

observed fluxes:

     1) The five algorithms were forced by half-hourly meteorological data ($U$, $\theta_a$, $\theta_w$, relative humidity, air pressure). Missing values were assigned for simulated heat fluxes when any observed values of $U$, $\theta_a$, $\theta_w$, and relative humidity were not available or when lake ice was present at a site.

2) Temporal averaging was applied to simulated and observed fluxes. We first calculated daily averaged $\lambda E$ and $H$. Gap-matching was applied to the simulated and observed fluxes. If either of the simulated or observed $\lambda E$ ($H$) were missing values at a half-hourly time step, both the simulated and observed $\lambda E$ ($H$) at this time step were not used for daily averaging. This was conducted so that daily averages from the simulation (roughly

continuous in time) were adequately compared with those from the observations, which had more frequent gaps. When more than 24 out of 48 data points are missing in a day, a missing value was assigned. For time series comparison, a 10-day moving average was applied to simulated and observed fluxes in order to smooth the synoptic variability and highlight comparison of the respective seasonal cycles. Daily averaging was used for one-

to-one comparisons (i.e. scatter plots).

     3) Root-mean-square errors (RMSEs) and mean bias were calculated for daily sensible and latent heat fluxes.

     4) Errors of daily $\lambda E$ and $H$ were calculated as functions of $\theta_w$-$\theta_a$, $q_w$-$q_a$, $U$, $C_{H,E}$, and $\zeta$.



### 3. Results

#### 3.1. Observed and modeled seasonal cycles

Figure 3 shows the time series of air temperature, water surface temperature from GLSEA, relative humidity, and wind speed at the four stations. The time series for Stannard Rock are

relatively GAP-FREE throughout the three years, while there are some data gaps in the time series for the other stations. The air temperature time series are characterized by a typical seasonal cycle, along with relatively warm and cold winters in 2012-2013 and 2013-2014, respectively (Fig. 3a). This is also reflected in the water surface temperature time series (Fig. 3b), where only White Shoal and Long Point were affected by ice cover in the winter (January-March) of 2012-2013,

shown as gaps in the time series, whereas all four stations were affected by ice cover in the winter of 2013-2014. In the spring and summer of 2012 (April-September), the water surface temperature was anomalously warm compared with the same periods of 2013 and 2014 (particularly at Stannard Rock). Relative humidity generally fluctuated between 50% and 90% (Fig. 3c), while wind speed (Fig. 3d) shows a weak seasonal cycle of relatively high wind speeds during fall and

winter (October-March) and low wind speeds during spring and summer (April-September).

Figures 4-7 show visual comparisons of 10-day moving average time series of $\lambda E$ and $H$ at each of the four stations. Overall, all five algorithms simulated the general seasonal cycles of $\lambda E$ and $H$, including the observed high fluxes during fall and winter and low fluxes during summer and spring that is typical for large North American lakes (Blanken et al., 1997, 2000, 2011; Spence et

al., 2011). On the other hand, there are notable overestimations of $\lambda E$ and $H$ by the original algorithms, particularly at Stannard Rock (Fig. 4) in the fall ($\lambda E$) and winter ($H$). The Stannard Rock dataset is largely gap-free, showing most continuous timeseries of seasonal cycles in $\lambda E$ and $H$, aside from periods of high ice coverage during the cold winter of 2013-2014. A few additional data gaps also occurred, including late summer of 2012, a longer data gap during January-May

2014, and very short data gap during December 2013.

Late-fall (October-December) $\lambda E$ and $H$ were relatively low in 2012 (3-month averages 84 W m$^{-2}$ for $\lambda E$ and 55 W m$^{-2}$ for $H$) and high in 2013 (119 W m$^{-2}$ in $\lambda E$, 85 W m$^{-2}$ for $H$), indicating preconditioning of the following mild and severe winters, respectively. During spring and summer of both years (April-September), the observed $\lambda E$ and $H$ are much lower due to the cool lake





surface relative to the overlying air. The simulated $\lambda E$ and/or $H$ mostly reproduced these lower values, but also showed occasionally negative values (Fig. 4), such as during May 2012 and July 2014. During these periods, the air was predominantly warmer than the water surface (i.e $T_w$-$T_a <$ 0, Fig. 1), and specific humidity gradients were near zero during May 2012 and reversed (i.e., air-

to-water) during July 2014, forcing the algorithms to simulate near-zero and negative (i.e. downward) fluxes, respectively. However, the observed $\lambda E$ and $H$ fluxes remained close to zero, but slightly positive.

The forcing dataset for White Shoal (Fig. 3) is relatively gap-free as well, but there was an invalid data period before October 2013 for the latent heat flux ($\lambda E$), and data gaps in $H$ due to ice

cover (Fig. 5). White Shoal tends to be influenced by ice cover even in mild winters, since typical south-westerly winds push ice in Lake Michigan downwind, causing ice accumulation occurs in northern parts of the lake near White Shoal. As such, the $H$ data gap during the mild winter of 2012-2013 at White Shoal was due to ice cover. These observations also showed contrasting late-fall heat fluxes during the two years: three-month average $H$ was 40 W m$^{-2}$ during October-

December 2012 and 61 W m$^{-2}$ during October-December 2013. Some model underestimation of the sensible heat flux ($H$) occurred during July-September 2013 and June-October 2014.

The Spectacle Reef forcing dataset (Fig. 3) and flux dataset (Fig. 6) both contained a long gap from March 2012 to September 2013 due to electrical problems from lightning strikes. A data gap in $\lambda E$ and $H$ during January-March 2014 was due to ice cover, but unlike White Shoal, Spectacle

Reef is less affected by ice cover. This is because winds carry ice that forms nearshore toward east and into. offshore Lake Huron, keeping the area around the flux tower largely in open water. Indeed, the station was not affected by ice cover in the winter of 2012-2013 (based on the NIC data), but this period was included in the above-referenced long data gap due to lighting strikes.

The dataset at Long Point (Fig. 7) shows the largest number of data gaps due to due to the

additional filtering according to wind direction of 180°-315°, which included typical south-westerly winds in this region. The significant data gaps at Spectacle Reef and Long Point, therefore, do not allow us to compare the late-fall fluxes between the anomalous two years. However, for the purpose of the algorithm verification, the data at the two stations were still valuable, and forcing datasets were largely continuous (Fig. 3).

Also shown in Figs. 4-7 are model results using both the original and updated $z_{0\theta,q}$ parameterizations (eq. 9). The original results of LS87, C89, Z98L, and J99 showed overestimation





of $\lambda E$ and $H$ at Stannard Rock by anywhere from 33-50% for most of the algorithms to ~80% overestimation for Z98L (both $\lambda E$ and $H$) and LS87 ($\lambda E$) (Fig. 4, Table 2). These overestimations were particularly obvious during high flux events in fall and winter (October-March). The overestimation at Stannard Rock was significantly lessened to roughly 24-33% error by using the

updated $z_{0\theta,q}$ formula (eq. 9). This is consistent with the findings of Deacu et al. (2012), who showed the improvements o simulated latent and sensible heat flux simulation by updating the roughness length scale parameterization at Stannard Rock for the December 2008 simulation period. Similar improvements are noted at White Shoal (Fig. 5), Spectacle Reef (Fig. 6), and Long Point (Fig. 7).

**3.2. Comparison of daily mean fluxes**

While the 10-day running mean time series of $\lambda E$ and $H$ provided an effective way to illustrate the overall cycle (Figs. 4-7), abrupt changes in $\lambda E$ and $H$ often occur on daily timescales, caused by the passage of frontal systems and cold air outbreaks (Blanken et al., 2008). Thus, we further evaluated the performance of the various algorithms at daily timescales by means of scatter plots

of observed and modeled daily mean heat fluxes (Figs. 8 and 9). Data points of $\lambda E$ (Fig. 8) diverged more from the 1:1 line than $H$ (Fig. 9), showing both overestimated fluxes (at Stannard Rock and Long Point with Z98L) and underestimated fluxes (at Spectacle Reef). Overall, the updated $z_{0\theta,q}$ formula reduced simulated $\lambda E$, generally bringing the fluxes into better agreement with observations. An exception to this occurred for $\lambda E$ at Spectacle Reef, where the agreement became slightly worse with

the updated formulation. The error reduction ratio was negative and the mean bias was more negative with the updated formulation at this station (Table 2). For $H$ (Fig. 9, Table 3), notable overestimation was seen in the original J99, LS87, and Z98L, particularly at relatively large heat loss values (> ~300 W m$^{-2}$). At Stannard Rock, Spectacle Reef, and Long Point, this overestimation was significantly improved with the updated $z_{0\theta,q}$ formula. At White Shoal, however, the improvement was not as

significant, according to error reduction ratios (Table 3) that were ~50% or greater at Stannard Rock, Spectacle Reef, and Long Point but only ~28% at White Shoal.

Stannard Rock showed small groups of $\lambda E$ and $H$ around the origin, where the simulated fluxes underestimated the observed fluxes (i.e. below the 1:1 line). The represented two summer periods when the observed fluxes were near zero, but the simulated fluxes were negative (see the discussion in section





3.1). At White Shoal, there was a population of $H$ values below the 1:1 line, representing periods when the simulation results underestimated the observations during July-September 2013 and June-October 2014 (see the discussion in section 3.1).

## 3.3. Error dependence on meteorological conditions and transfer coefficients

Figures 10 and 11 show the magnitude of error in simulated daily $\lambda E$ and $H$ (i.e., difference from observations) as functions of $T_w$-$T_a$, $q_w$-$q_a$, $C_H$, $C_E$, $U$, and $\zeta$=$z/L$ for the five algorithms at Stannard Rock. Similar results were observed in the error and bias analyses at the other sites (supplementary Figures S1-S6). There are several features common in all the algorithms: the $H$ ($\lambda E$) errors were positively correlated with $T_w$-$T_a$ ($q_w$-$q_a$) for negative values of $T_w$-$T_a$ ($q_w$-$q_a$); the amplitudes of the errors become large (both positive and negative) as wind speed increases; and the majority of data are in the range -2<$\zeta$<0 (unstable). Most notably, the transfer coefficients $C_H$ and $C_E$ were significantly reduced with the updated $z_{0\theta,q}$ formula, which also reduced the error in the $\lambda E$ and $H$ simulations. This was to be expected, since $z_{0\theta}$ and $z_{0q}$ are directly translated into $C_H$ and $C_E$ respectively. Although Figs. 10 and 11 do not clearly show the data density, $C_H$ and $C_E$ were concentrated around 0.0012, which was roughly a neutral value (Kantha and Clayson, 2000b; Smith, 1989). The study period did not include the occurrence of highly unstable conditions ($\zeta$<<-1, Figs. 10 and 11, Supplemental figures S1-S6). Therefore, the period was not sufficient to evaluate the convective behaviour treatment in COARE. Also, the study period did not include sustainedlow wind speeds. Fairall et al. (1996a) note that the gustiness parameterization has only a modest effect until the wind speed becomes less than 2-3 m s$^{-1}$. The wind speeds during our study period were mostly greater than 3 m s$^{-2}$ (Figs. 10 and 11, Supplemental figures S1-S6) and therefore did not allow us to evaluate the influence of the gustiness parameterizations in COARE and Z98L.

## 4. Discussion

The simulation results of four of the five algorithms investigated here (J99, C89, LS87, Z98L) were overall improved by the updated $z_{0\theta,q}$ formula (Eq. 9), bringing the simulation results into




closer correspondence with the COARE simulations. In our study period, the simulation results were less sensitive to the other factors in the algorithms: For example, we did not observe a clear difference in the results when using the various stability functions (i.e. $\Psi_{M,\theta,q}$) in the algorithms. As mentioned in 3.3, our study period did not have conditions conductive to evaluating the

convective behaviour treatment in COARE and the gustiness effect in COARE and Z98L. Thus, the most sensitive factor in our analyses was the parameterization of roughness length scales for temperature and humidity ($z_{0\theta}$ and $z_{0q}$). Formulae for $z_{0\theta,q}$ with smooth surface parameterization (such as Eq. 9) have been widely used for air-sea interaction modeling (e.g. Beljaars, 1994; Fairall et al., 2003) and have been also verified in lake applications (Deacu et al., 2012). It is reasonable

that future updates of the four algorithms should include the updated or similar formulation of $z_{0\theta,q}$

On the other hand, the inclusion of the updated $z_{0\theta,q}$ formula will not guarantee immediate improvement of the parent model systems. This is because each of the model systems is complex and must embrace uncertainties from all aspects, including forcing, dynamics, and boundary conditions. Typically, such a system is calibrated to provide best estimates of certain variables for

its own purpose (e.g. water temperature for the implementation of FVCOM in GLOFS), and a sudden change to a single aspect of the system would lose a balance that has been achieved by extensive calibration. An ideal approach to improve model systems would have to be more comprehensive.

The reasons for the discrepancy between the simulated (negative) and observed (near zero, but

positive) $\lambda E$ and $H$ values during summer at Stannard Rock (Fig. 4, around May 2012 and July 2014) are not fully understood. A similar discrepancy was found at Long Point (Fig. 7, around April 2013) and at White Shoal (Fig. 5, around June 2014) although the discrepancy was only for $H$ and the magnitude of the discrepancy was smaller than at Stannard Rock. The discrepancy remained even after updating the $z_{0\theta,q}$ formula. During these periods, the temperature gradients

between the air (at sensor heights) and at the water surface were commonly negative (the air was warmer) and wind speeds ranged from 6-12 m s$^{-1}$., resulting in the negative fluxes (i.e., downward) simulated by the bulk flux algorithms. One possible reason is that the sensors were above the constant flux layer during these periods, and therefore, the similarity theory on which the bulk flux algorithms are based on was not applicable. However, given the sensor heights at these stations,

such a condition would be rare and there is no way to verify this possibility.





Other possible sources of the discrepancy would be in the forcing data. Particularly, uncertainties in the water surface temperature data, GLSEA, should be noted. As described in section 2.1.2, the information for cloudy areas is created using an interpolation method from the satellite imageries within ±10 days. Therefore, the GLSEA data tends to have lower accuracy and could miss abrupt changes in water surface temperature for cloudy days. The IRT-measured water surface temperature showed somewhat warmer water surface temperature than the GLSEA data during these discrepancy periods (Supplemental figure S7), indicating possible underestimation of water surface temperature in the GLSEA data (i.e. false negative $\lambda E$ and $H$). However, we concluded earlier that the accuracy of the IRT-measured water surface temperature was limited (see section 2.1.2). An ideal way to confirm the GLSEA accuracy for such analyses would be an *in situ* water surface temperature measurements at the flux tower sites using buoys, for example (which is now being done at Stannard Rock and planned for White Shoal).

The normalized RMSEs at Long Point were worse than those at the other stations even though we filtered out the data with wind directions in the range of $180^o$-$315^o$ were filtered out. This filtering window should be sufficiently large to remove the land surface contamination. We again suspect the water surface temperature data could be an error source. As noted in section 2.1, the station is on the shore of a narrow peninsula extending into Lake Erie. The satellite-based observations of water surface temperature tend to lose their accuracy near the coast, and thus the GLSEA accuracy at this station could be lower. For such a location, FVCOM, a full hydrodynamic model, may be appropriate to reproduce the observed fluxes, but it should have sufficient horizontal resolution to represent the complex bathymetry around the peninsula, which is essential to reproduce the spatial pattern of heat capacity in the water column correctly, and therefore, the water surface temperature.

## 5. Conclusions

This study focused on the validation of surface latent and sensible heat fluxes ($\lambda E$ and $H$) from the surface of the Great Lakes. We isolated the surface flux algorithms commonly used in Great Lakes physical modeling and tested each algorithm using observed meteorology and lake surface temperatures by comparing their output to several eddy covariance stations within the GLEN that





measure *in situ* lake surface fluxes. All algorithms reproduced the seasonal cycle of $\lambda E$ and $H$ reasonably well during a warm period (2012-mid 2013) and cold period (late 2013-2014). However, four of the original algorithms (i.e., except for COARE) presented notable disagreement with the observations under certain conditions; significant positive biases in $H$ were found under

high upward heat flux conditions for the algorithms other than COARE; the errors in $H$ were also positively correlated with the temperature difference between air and water.

These errors were significantly improved by introducing the updated $z_{0\theta,q}$ formula based on Fairall et al., (2003), which is well supported by the air-sea interaction modeling community. The update led to reduced transfer coefficients $C_H$ and $C_E$, reducing the overestimation of the simulated

heat fluxes. While it is reasonable to adopt the updated formula in the parent model systems where these algorithms are included, this does not guarantee immediate improvement of simulations by the parent model systems, since these model systems are calibrated to provide best simulations for certain variables by embracing uncertainties in all aspects. We used *in situ* meteorological forcing to drive the algorithms, which is generally ideal, but in operational practice, it is not possible to

use *in situ* data over the entire lake surface. For example, GLOFS uses interpolated and/or model-forecasted meteorological forcing, which inevitably includes additional sources of error.

We successfully evaluated the flux algorithms, which are an important aspect of Great Lakes water and energy balance modeling, and we identified and reduced errors in simulated heat fluxes from these algorithms. We recommend that bulk flux algorithms use an appropriate

parameterization for $z_{0\theta}$ and $z_{0q}$ instead of assuming them equal to $z_0$, or simply employ the COARE algorithm, which presented the best agreement with the eddy covariance measurements in this study. We also recommend simultaneous *in situ* measurement of water surface temperature at the flux tower locations in order to allow more robust comparison between the eddy covariance measurements and simulated $\lambda E$ and $H$ by a column model (e.g., the five algorithms independently

driven by the forcing data in this study).

Accurate simulation of the turbulent heat fluxes from the lake surface is important to a wide range of lake-atmosphere and earth system applications, from long-term water balance estimates to numerical prediction of lake levels, weather, lake ice, and regional climate. Communities are increasingly dependent on numerical geophysical models for these types of applications. The

continued monitoring of turbulent heat fluxes at the offshore GLEN sites is critical for such models to be improved in future studies.





**Acknowledgments**

This research is funded by the US Environmental Protection Agency's Great Lakes Restoration Initiative (GLRI) and the NOAA' Coastal Storms Program awarded to the Cooperative Institute for Great Lakes Research (CIGLR) through the NOAA Cooperative Agreement with the University of Michigan (NA12OAR4320071). Dr. Charusombat was supported by a National Research Council Research Associateship award at the NOAA Great Lakes Environmental Research Laboratory. The data used in this research were kindly provided by the Great Lakes Evaporation Network (GLEN). GLEN data compilation and publication were provided by LimnoTech, the University of Colorado at Boulder, and Environment and Climate Change Canada under Award/Contract No. 10042-400759 from the International Joint Commission (IJC) through a subcontract with the Great Lakes Observing System (GLOS). The statements, findings, conclusions, and recommendations are those of the authors and do not reflect the views of GLEN, LimnoTech, the University of Colorado at Boulder, Environment and Climate Change Canada, the IJC, or GLOS. The authors thank Drs. Chris Fairall and Dev Niyogi for comments that improved the quality of this manuscript. This is GLERL Contribution Number XXXX and CIGLR contribution XXXX.

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



**Table 1 – Summary of flux algorithm specifications.**

| Algorithm name | Parent Model | Stability | | $z_0$ | $z_{0\theta,q}$ | Gustiness | References |
|---|---|---|---|---|---|---|---|
| | | Unstable | Stable | | | No | |
| LS87 | FVCOM | Similar to Businger et al. (1971) | Holtslag et al. (1990) | $\alpha \dfrac{u^{*2}}{g} + 0.11 \dfrac{\nu}{u^*}$ $\alpha$=0.011 | Assume equal to $z_0$. | No | Liu and Schwab (1987) |
| C89 | LLTM | Businger et al. (1971) | Holtslag et al. (1990) | $\alpha \dfrac{u^{*2}}{g}$ $\alpha$=0.0101 | Assume equal to $z_0$. | No | Croley, 1989a,b) |
| Z98L | WRF-Lake | Businger et al. (1971) | Holtslag et al. (1990) | 0.001 m (Smith, 1988 for ocean) | Assume equal to $z_0$. (Brutsaert, 1975 for ocean) | Fairall et al. (1996), $\beta$=1.0 | Zeng (1998) |
| J99 | FVCOM, UG-CICE | Businger et al. (1971) | Beljaars and Holtslag, (1991) | $z\exp\left[-\kappa\left(\dfrac{2.7\times10^{-3}}{U} + 1.42\times10^{-4} + 7.64 \times10^{-5}U\right)^{-1}\right]$ (Large et al., 1994) | Assume equal to $z_0$. (Jordan (1999) used Andreas, (1987) for ice surface. | No | Jordan (1999), Hunke et al. (2015) |
| COARE | FVCOM | Businger et al. (1971), Convective behavior: Fairall et al.,(1996) | Beljaars and Holtslag, (1991) | $\alpha \dfrac{u^{*2}}{g} + 0.11 \dfrac{\nu}{u^*}$ $\alpha$: function of wind speed | min (1.6 $\times10^{-4}$, 5.8 $\times10^{-5}Rr^{-0.72}$) | Fairall et al. (1996), $\beta$=1.2 | Fairall et al., (1996ab), Edson et al. (2013) |

**Table 2** Statistics of simulated latent heat flux $\lambda E$ for 2012-2014. For J99, LS87, Z98L, and C89, RMSEs with the updated $z_{0\theta,q}$ formulation are shown. Numbers in parentheses denote RMSEs with the original $z_{0T,q}$ formulation. An error reduction ratio (%) is calculated for mean RMSEs of J99, LS87, Z98L, and C89. A mean flux (W m$^{-2}$) and mean normalized RMSE are calculated for all the five algorithms.

| | RMSE [Wm$^{-2}$] | | | | | Error reduction ratio [%] | Mean flux [W m$^{-2}$] | Mean Normalized RMSE | Mean bias [%] |
|---|---|---|---|---|---|---|---|---|---|
| | COARE | J99 | LS87 | Z98L | C89 | | | | |
| Stannard Rock | 27.2 | 33.1 (32.6) | 29.3 (39.9) | 29.1 (79.9) | 29.1 (38.9) | 37.0 | 56.9 | 0.53 (0.84) | 29.6 (38.3) |
| White Shoal | 25.2 | 36. (25.3) | 28.3 (25.4) | 27.8 (68.0) | 27.6 (25.8) | 17.0 | 61.1 | 0.49 (0.59) | 1.4 (24.0) |
| Spectacle Reef | 70.4 | 83.8 (66.8) | 68.5 (61.9) | 67.4 (72.6) | 71.3 (62.5) | -10.3 | 116.1 | 0.63 (0.57) | -27.8 (-3.2) |
| Long Point | 42.9 | 40.1 (42.1) | 47.9 (46.5) | 49.1 (104.3) | 45.8 (47.8) | 24.1 | 50.7 | 0.90 (1.19) | 27.4 (49.6) |
| Mean RMSE [Wm$^{-2}$] | 41.4 | 48.3 (41.7) | 43.5 (43.5) | 43.3 (81.2) | 43.5 (43.8) | 15.0 | 81.5 | 0.52 (0.64) | - |
| Mean bias [%] | 4.3 | -9.1 (8.7) | 15.7 (18.9) | 16.5 (88.5) | 10.9 (19.6) | - | - | - | 7.7 (27.1) |

**Table 3** Same as Table 2, but for sensible heat flux $H$.

| | RMSE [Wm$^{-2}$] | | | | | Error reduction ratio [%] | Mean flux [W m$^{-2}$] | Normalized RMSE | Mean bias [%] |
|---|---|---|---|---|---|---|---|---|---|
| | COARE | J99 | LS87 | Z98L | C89 | | | | |
| Stannard Rock | 27.4 | 29.1 (50.2) | 26.3 (84.7) | 26.5 (77.4) | 23.6 (32.1) | 56.8 | 39.1 | 0.68 (1.56) | 26.6 (48.9) |
| White Shoal | 32.3 | 31.4 (37.9) | 31.8 (50.8) | 31.9 (52.8) | 31.0 (32.9) | 27.7 | 40.7 | 0.78 (1.07) | -24.9 (7.8) |
| Spectacle Reef | 11.4 | 13.2 (27.2) | 13.9 (60.4) | 11.9 (65.3) | 13.3 (13.8) | 68.6 | 46.1 | 0.28 (0.90) | 6.3 (44.8) |
| Long Point | 27.2 | 26.7 (45.5) | 28.5 (65.6) | 27.6 (63.2) | 21.5 (32.9) | 49.7 | 11.7 | 2.2 (4.4) | 18.5 (31.4) |
| Mean RMSE [Wm$^{-2}$] | 24.6 | 25.1 (40.2) | 25.1 (65.4) | 24.5 (64.7) | 22.4 (28.0) | 51.0 | 38.0 | 0.64 (1.30) | - |
| Mean bias [%] | 4.2 | 7.5 (31.4) | 13.6 (61.7) | 6.3 (62.0) | 1.5 (11.0) | - | - | - | 6.6 (33.2) |



**Captions of Figures**

**Figure 1. Map of the Laurentian Great Lakes including location of monitoring stations used in this study. Adapted from Lenters et al. (2013). Instrument heights above the mean water level are 32.5 m at Stannard Rock, 29.5 m at Long Point, 30.0 m at Spectacle Reef, and 42.8 m at White Shoal.**

**Figure 2. Schematic diagram showing the relationship between the parent model systems (FVCOM, WRF-Lake, and LLTM) and the flux algorithms used in the parent model systems. Detail description of each flux algorithm is listed in Table 1.**

**Figure 3. 10-day running mean time series of meteorological variables at the four stations. Air temperature and relative humidity were measured with Vaisala HMP45C thermohygrometers and wind speed were measured with the CSAT-3 (See section 2.1.1 or Figure 1 for the sensor heights). Water surface temperature is taken from GLSEA. Data at pixels closest to the stations are used. The data gaps in water surface temperature from January to April denote periods during which the site was affected by lake ice cover. Measurements at Long Point and White Shoal started in May and June of 2012. There is also a long data gap between February 2012 and June 2013 at Spectacle Reef.**

**Figure 4. 10-day running mean time series of latent ($\lambda E$) and sensible ($H$) heat fluxes at Stannard Rock. Black lines denote observed $\lambda E$ and $H$ and the same for (a), (b) and (c), (d), respectively. The $\lambda E$ and $H$ simulations employ the original $z_{0\theta,q}$ formula in (a), (c) and with the updated $z_{0\theta,q}$ formula in (b) and (d). The COARE simulation results are unchanged from (a) to (b) or from (c) to (d).**

**Figure 5. The same as Figure 4, but at White Shoal.**

**Figure 7. The same as Figure 4, but at Long Point.**

**Figure 6. The same as Figure 4, but at Spectacle Reef.**

**Figure 8. Scatter plots of latent heat flux ($\lambda E$) comparing the observed ($x$-axis) and the simulated ($y$-axis) daily mean fluxes. Each row shows comparisons with a specific**




algorithm at the four stations, while each column shows comparisons with the five algorithms at a specific station. Grey and blue dots indicate the results with the original and updated $z_{0\theta,q}$ formulae, respectively.

**Figure 9.** The same as Figure 8, but for sensible heat flux (***H***).

**Figure 10.** Errors in daily mean latent heat flux (***y***-axis) versus specific humidity difference between the water surface and air at the sensor height $q_w$-$q_a$ [kg kg⁻¹], transfer coefficient $C_E$ [-], wind speed ***U*** [m s⁻¹], and stability factor ***z/L*** (***x***-axis) for the five algorithms at Stannard Rock. Grey and blue dots indicate the results using the original and updated $z_{0\theta,q}$ formulae, respectively.

**Figure 11.** Errors in daily mean sensible heat flux (***y***-axis) versus temperature difference between the water surface and air at the sensor height $\theta_w$- $\theta_a$ [°C], transfer coefficient $C_H$ [-], wind speed ***U*** [m s⁻¹], and stability factor ***z/L*** (***x***-axis) for the five algorithms at Stannard Rock. Grey and blue dots indicate the results with the original and updated $z_{0\theta,q}$ formulae, respectively.





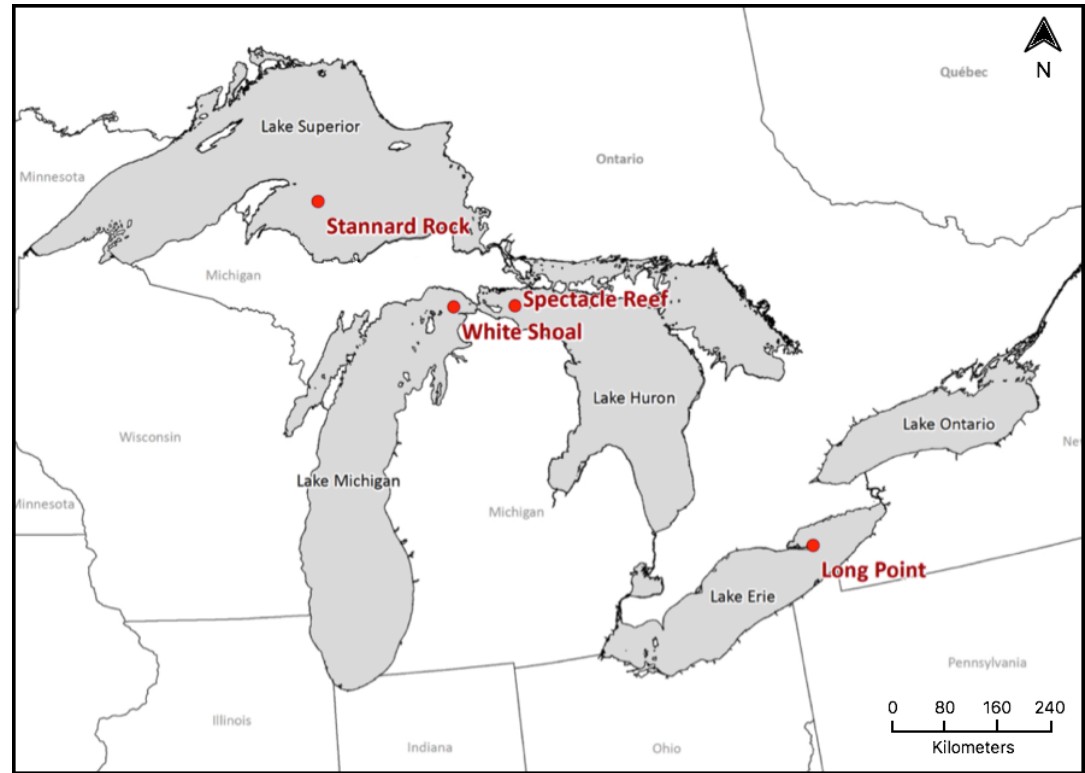

**Figure 1. Map of the Laurentian Great Lakes including location of monitoring stations used in this study. Adapted from Lenters et al. (2013). Instrument heights above the mean water level are 32.5 m at Stannard Rock, 29.5 m at Long Point, 30.0 m at Spectacle Reef, and 42.8 m at White Shoal.**





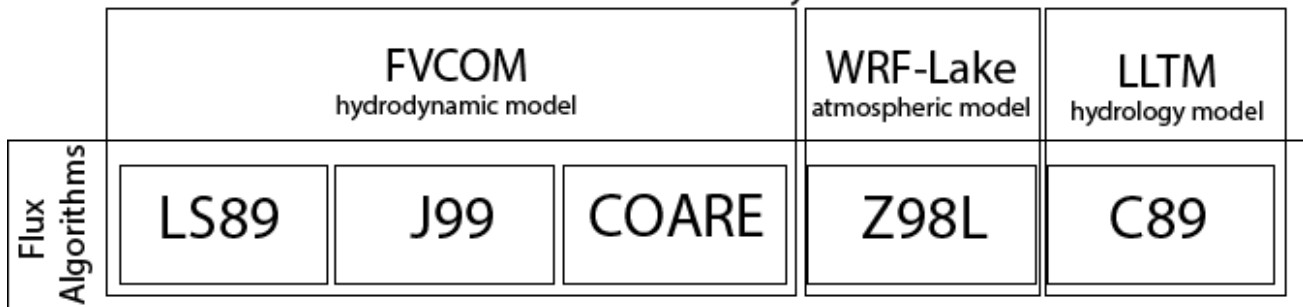

**Figure 2. Schematic diagram showing the relationship between the parent model systems (FVCOM, WRF-Lake, and LLTM) and the flux algorithms used in the parent model systems. Detail description of each flux algorithm is listed in Table 1.**





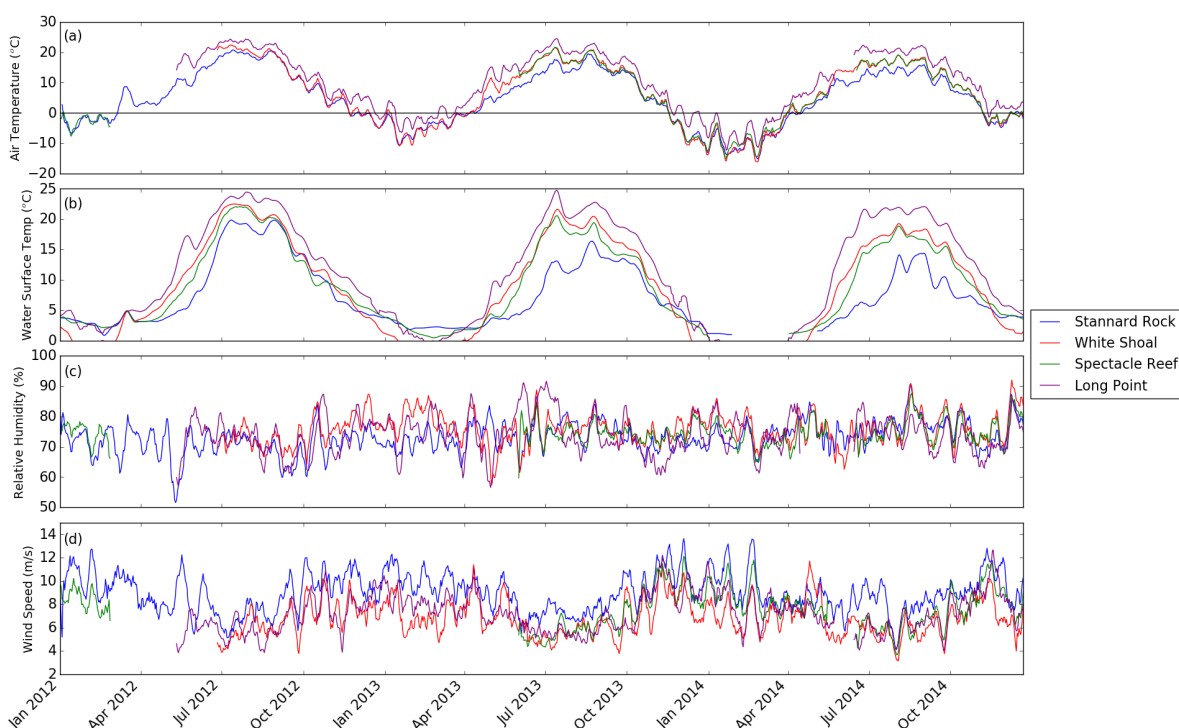

**Figure 3. 10-day running mean time series of meteorological variables at the four stations. Air temperature and relative humidity were measured with Vaisala HMP45C thermohygrometers and wind speed were measured with the CSAT-3 (See section 2.1.1 or Figure 1 for the sensor heights). Water surface temperature is taken from GLSEA. Data at pixels closest to the stations are used. The data gaps in water surface temperature from January to April denote periods during which the site was affected by lake ice cover. Measurements at Long Point and White Shoal started in May and June of 2012. There is also a long data gap between February 2012 and June 2013 at Spectacle Reef.**

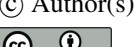


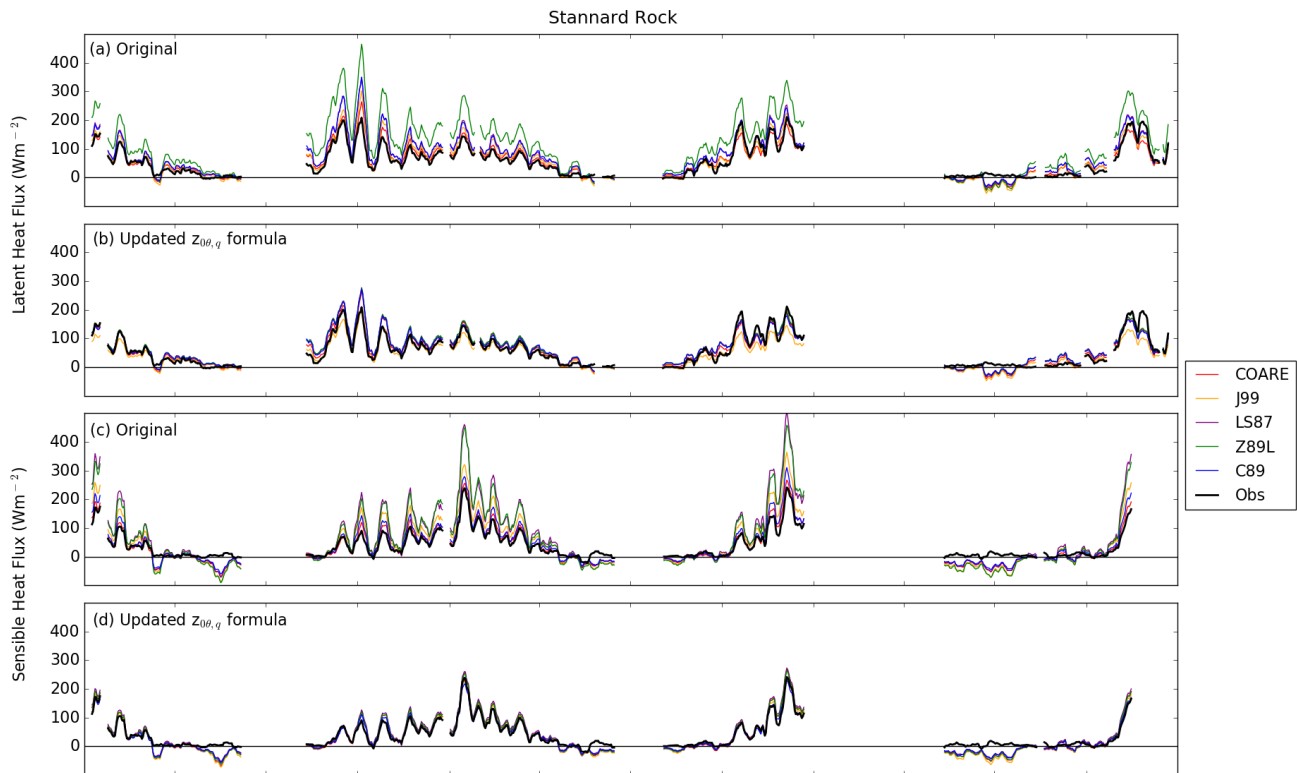

**Figure 4. 10-day running mean time series of latent ($\lambda E$) and sensible ($H$) heat fluxes at Stannard Rock. Black lines denote observed $\lambda E$ and $H$ and the same for (a), (b) and (c), (d), respectively. The $\lambda E$ and $H$ simulations employ the original $z_{0\theta,q}$ formula in (a), (c) and with the updated $z_{0\theta,q}$ formula in (b) and (d). The COARE simulation results are unchanged from (a) to (b) or from (c) to (d).**





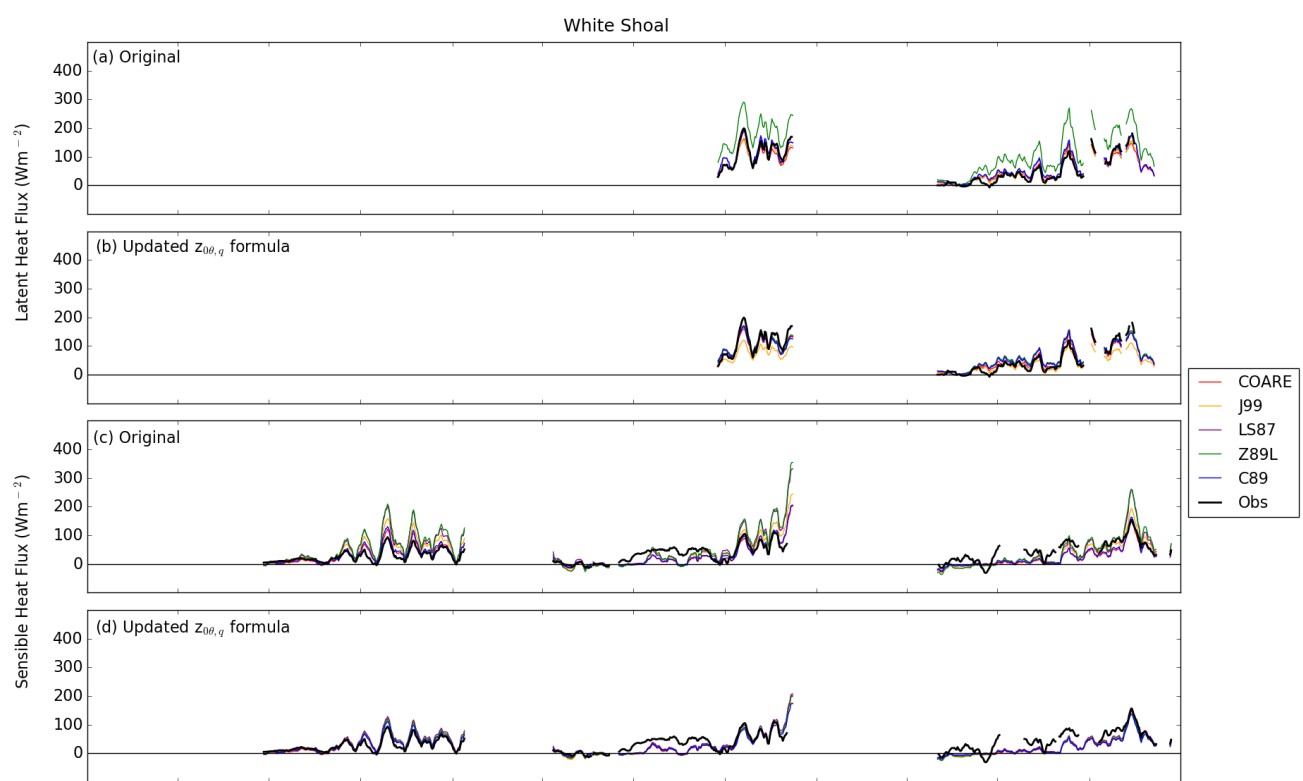

**Figure 5. The same as Figure 4, but at White Shoal.**

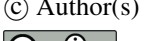



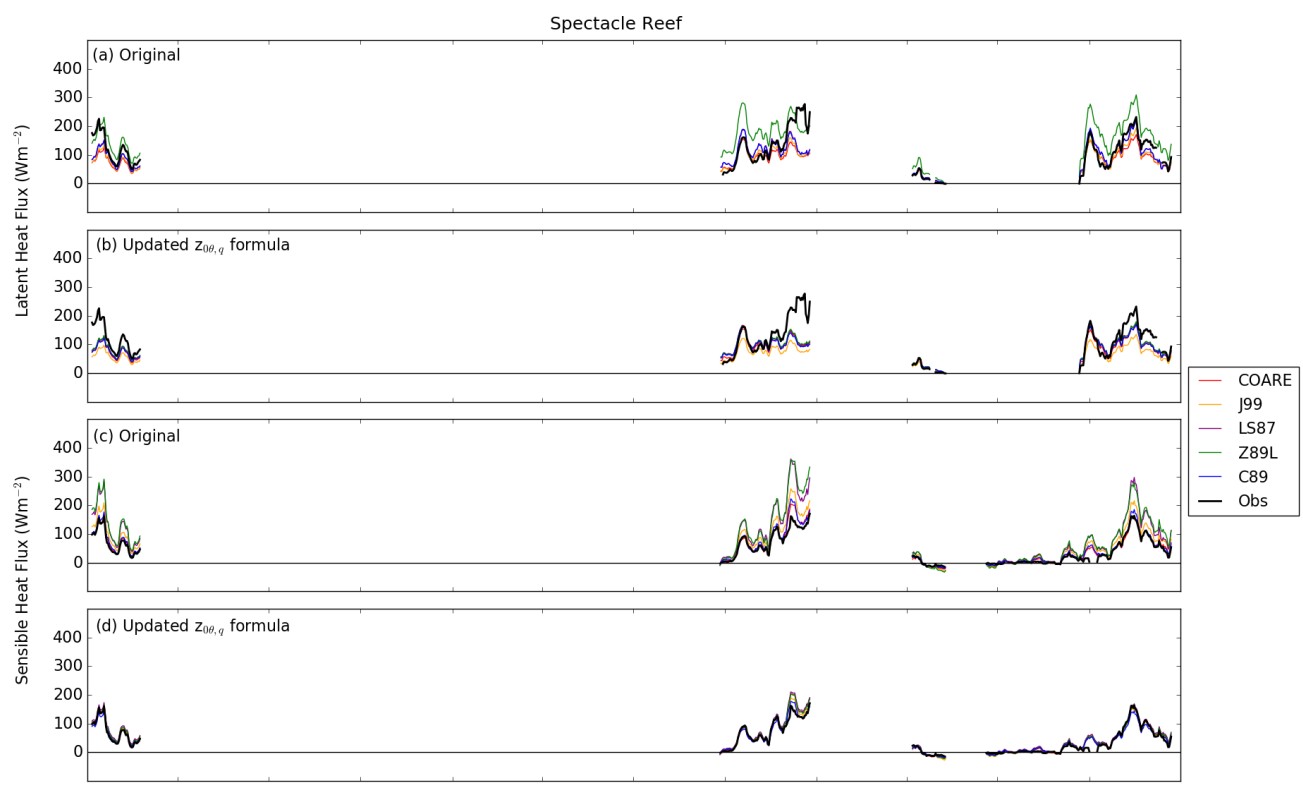

Figure 6. The same as Figure 4, but at Spectacle Reef.





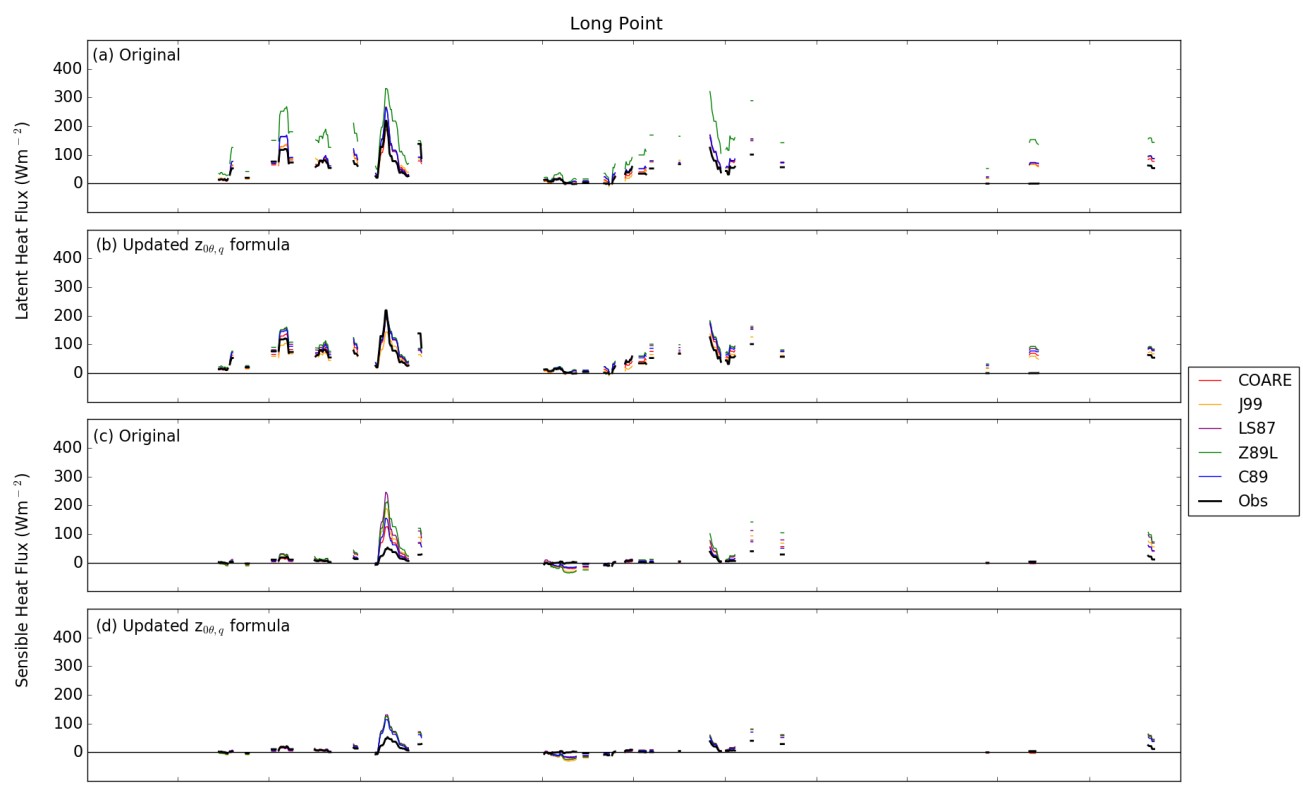

**Figure 7. The same as Figure 4, but at Long Point.**





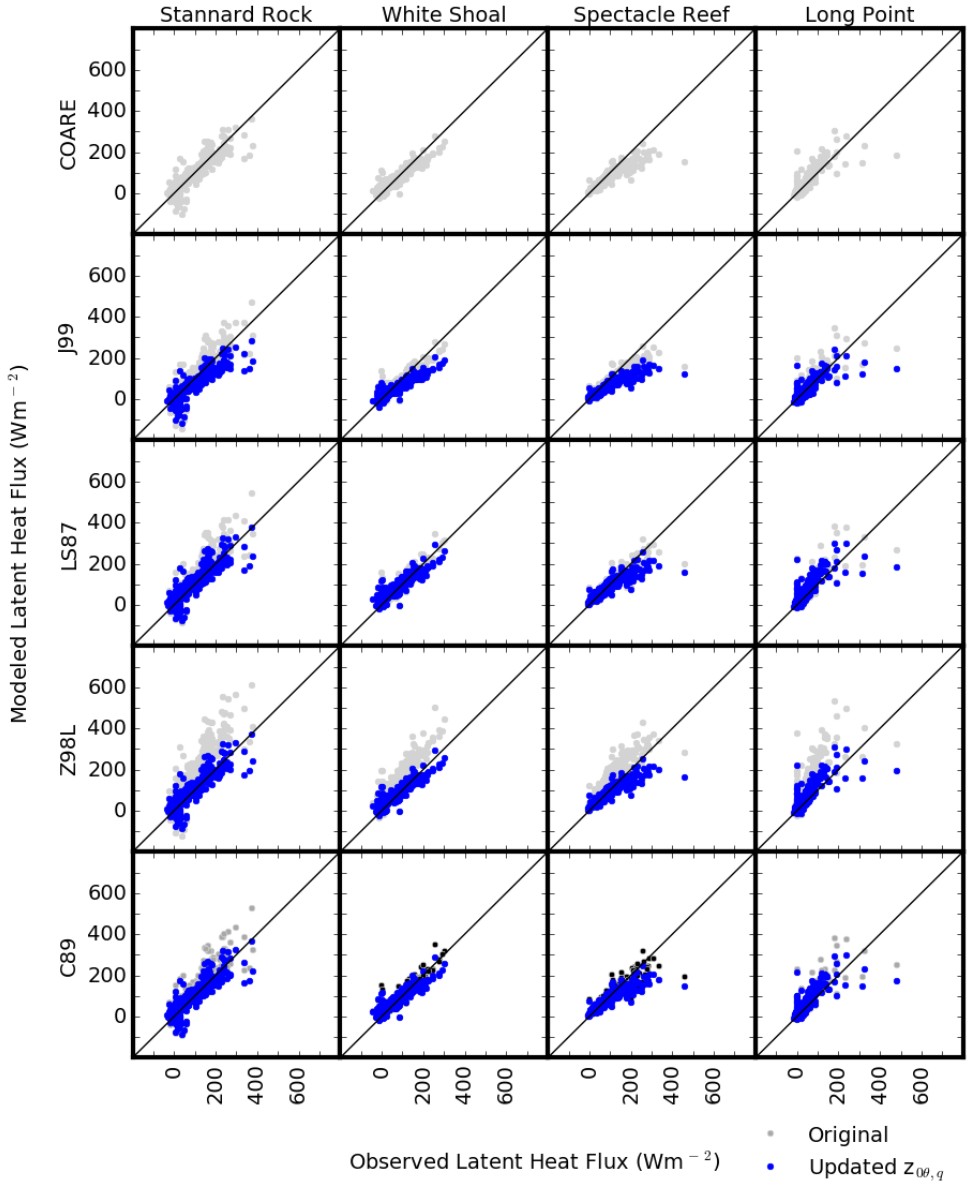

**Figure 8. Scatter plots of latent heat flux ($\lambda E$) comparing the observed (*x*-axis) and the simulated (*y*-axis) daily mean fluxes. Each row shows comparisons with a specific algorithm at the four stations, while each column shows comparisons with the five algorithms at a specific station. Grey and blue dots indicate the results with the original and updated $z_{0\theta,q}$ formulae, respectively.**



**Figure 9. The same as Figure 8, but for sensible heat flux (*H*).**





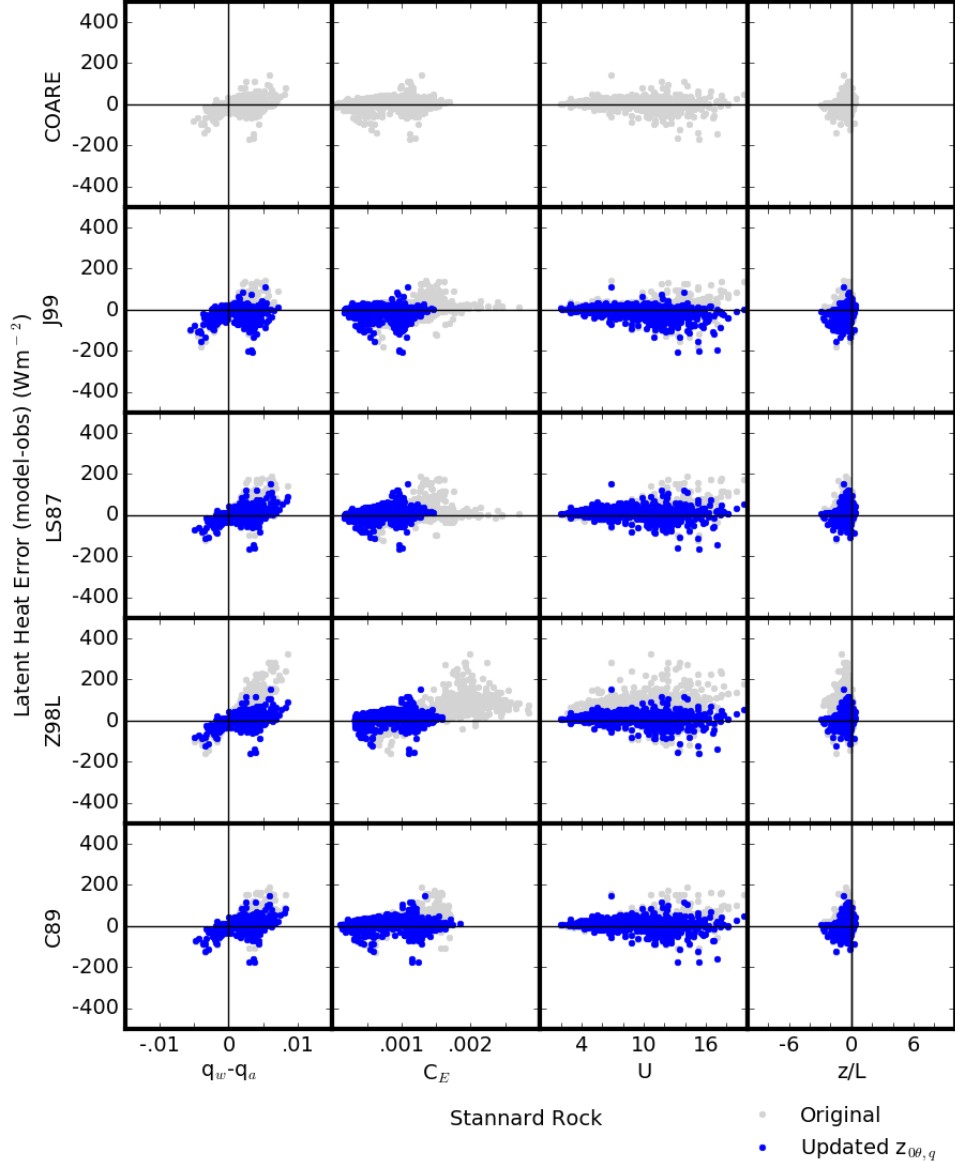

**Figure 10. Errors in daily mean latent heat flux (*y*-axis) versus specific humidity difference between the water surface and air at the sensor height $q_w$-$q_a$ [kg kg$^{-1}$], transfer coefficient $C_E$ [-], wind speed *U* [m s$^{-1}$], and stability factor *z/L* (*x*-axis) for the five algorithms at Stannard Rock. Grey and blue dots indicate the results using the original and updated $z_{0\theta,q}$ formulae, respectively.**





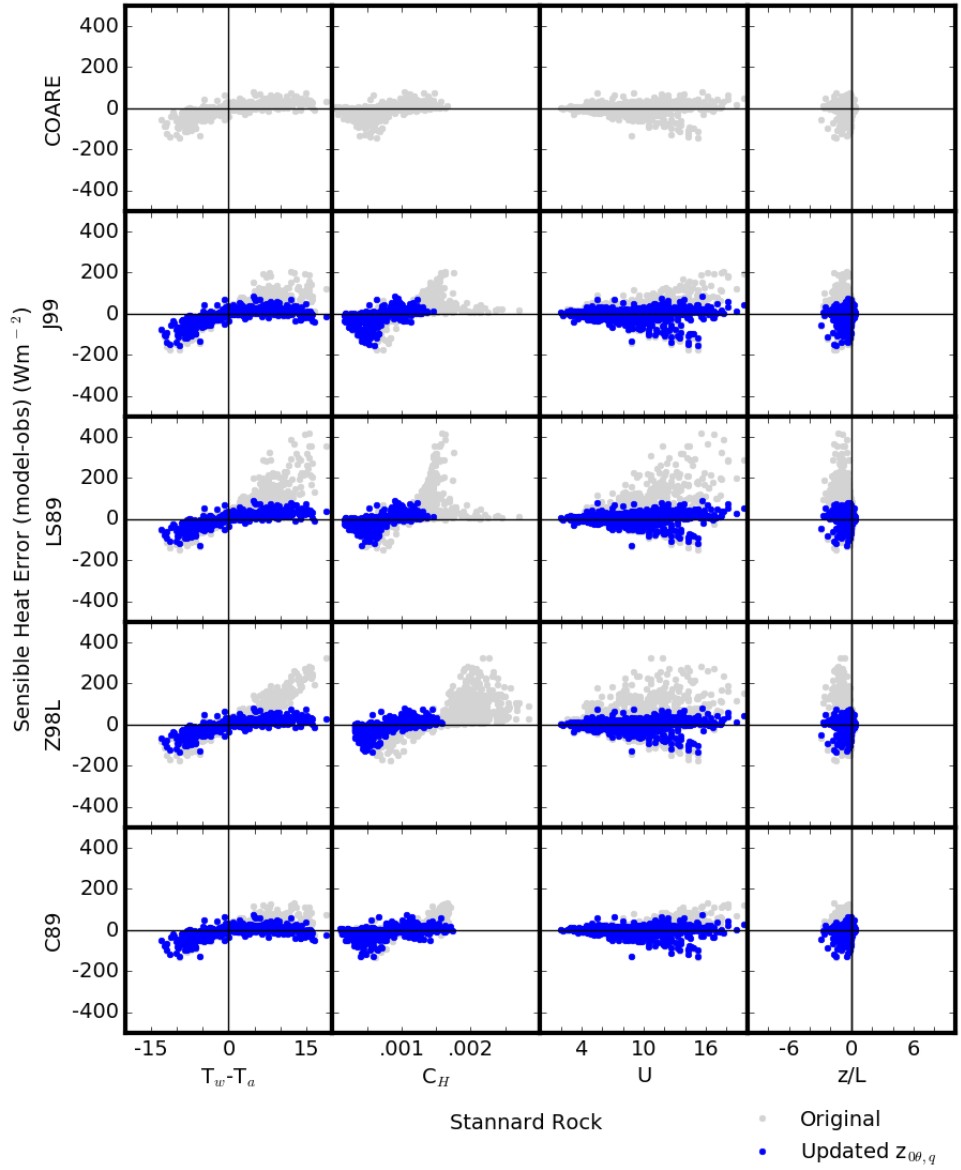

**Figure 11. Errors in daily mean sensible heat flux (*y*-axis) versus temperature difference between the water surface and air at the sensor height $\theta_w - \theta_a$ [°C], transfer coefficient $C_H$ [-], wind speed $U$ [m s$^{-1}$], and stability factor $z/L$ (*x*-axis) for the five algorithms at Stannard Rock. Grey and blue dots indicate the results with the original and updated $z_{0\theta,q}$ formulae, respectively.**

