# Peer review of "Evaluating and improving modeled turbulent heat fluxes across the North American Great Lakes"

_Hydrology and Earth System Sciences, 2017_

## Short Comment (SC1) · 17 Jan 2018

Dear Editor and Authors,

I'd like to leave some comments on this interesting study. In my opinion, its most obvious weakness is to use the satellite-derived water surface temperature to simulate sensible and latent heat fluxes. As water temperature is a sensitive variable for these simulations, the use of satellite-derived data for site-level applications may introduce large uncertainties, especially during the bad weather seasons when more data become unreliable. Second, the use of daily water temperature and half-hourly meteorological variables simultaneously may also introduce additional errors, for example,

during deep water mixing when energy balance is fast approached.

Also, the authors mention that the flux algorithm (Z98L) of the WRF model is adapted from the CLM 4.5 where Z98L assumes that the roughness length scale of momentum, $z0$, is a constant 0.001 m and the roughness length scales of momentum, temperature and humidity are the same. Actually, in the lake model of CLM4.5 (CLM-CLISSS), $z0$ is updated dynamically and three roughness length scales are not equal (for frozen lakes, $z0$ is fixed). Please check with Subin et al. (2012) for accuracy.

Z. M.Subin, W. J.Riley and D.Mironov (2012), An improved lake model for climate simulations: Model structure, evaluation, and sensitivity analyses in CESM1, J. Adv. Model. Earth Syst., 4, M02001, doi:10.1029/2011MS000072.

Best regards

Zeli Tan, Ph.D Pacific Northwest National Laboratory

---

## Author Comment (AC1) · 24 Jan 2018

Indeed, using the satellite-derived water surface temperature (GLSEA) would be the most significant source of simulation errors. As described in Section 2.1.2, it is a composite analysis using cloud-free portions of satellite imageries and therefore the accuracy tends to be lower on cloudy days (discussed in Section 4). However, we confirmed that the GLSEA water surface temperature agreed reasonably with offshore buoy measurements during these periods. Even though offshore buoy sites are not collocated with the stations for eddy covariance measurements and buoy data are typically not available during December-March (mainly due to ice cover), the agreement

between the GLSEA water surface temperature and buoy measurements provides us some confidence in the accuracy in the GLSEA water surface temperature. A recent study by Moukomla and Blanken (2016) tested an experimental method to derive water surface temperature from MODIS (or Moderate Resolution Imaging Spectroradiometer) for all-sky conditions. Such a product may also be tested in the future. This point will be added in the revision process. However, as described in Section 4, in-situ measurements (e.g. thermistor) are desired for a more reliable simulation.

As for the concern in using daily water surface temperature along with half-hourly meteorological variables, the only hydrodynamic process that can cause rapid and subdaily changes in the water surface temperature is upwelling, which are readily found in nearshore waters but not at the offshore locations of the eddy covariance stations. If the commenter is referring to deepwater mixing during fall overturn, that generally does not cause rapid temperature changes that would require sub-daily resolution. Therefore, daily water temperature would be sufficient for this study.

As for the note on the roughness length scale treatment in CLM4.5, this indeed needs to be mentioned in the manuscript so that it clarifies in CLM4.5 $z_0$ is updated dynamically and is not equal to $z_{0T,q}$. On the other hand, in the WRF application, at least in the latest version of 3.9.1, $z_0$ is still constant 0.001 m for unfrozen lake and $z_{0T,q}$ are set as equal to $z_0$. In the revision process, we will make sure that these points are made clear.

Moukomla, S. and Blanken, P. D.: Remote sensing of the North American Laurentian Great Lakes' surface temperature, Remote Sens., 8(4), 1–15, doi:10.3390/rs8040286, 2016.

---

## Referee Comment (RC1) · Anonymous Referee #1 · 11 Feb 2018

This is generally a well written paper with a few minor grammatical errors. I have listed these below. My main issue is with the conclusions which read like a summary rather than conclusions. The main conclusion of this paper are that when corrected for z0ïĄś,q the four models give similar results. I also wonder if the amount of measurement points that could be included would be helped by using the foot print analysis in McGloin et al. (2014, Water Resources Res., 50:494-513). This may allow more of the data at Long Point to be included.

Minor Corrections 1. Page 14, Line 11. Delete 'occurs'. 2. Page 14, lines 20-21. The sentence starting 'This is because. . .' does not make sense as written and has two

full stops in it. 3. Page 14, line 24. Delete the second 'due to'. 4. Page 15, line 6, Replace 'o' with 'of'. 5. Page 15, lines25-26. I suggest rewriting after 'significant' with (28% compared to 50% for Stannard Rock, Spectacle Reef, and Long Point according to error reduction ratios (Table 3). 6. Page 15, line 28. Replace 'The' with 'These data'. 7. Page 16, line 20. Space needed between 'sustained' and 'low'. 8. Page 17, line 17. 'more comprehensive.' in what? The authors need to elaborate how the models need to be improved and why. 9. Page 17, line 30. Insert 'from the present data set.' After 'possibility'. 10. Page 18, line 8. Replace '(i.e.' with 'and resulting in' and delete ')' after 'H'.
* * *

---

## Author Comment (AC2) · 23 Feb 2018

As for the referee's concern with the Conclusions section, we renamed the title of Section 5 to "Summary and Conclusions", since this is more appropriate, as the referee pointed out. Also, we added to this section a sentence "With the updated formula for $z_{0\theta,q}$, the four models (LS87, C89, J99, Z98L) simulated similar heat fluxes to COARE's"

The suggestion to use the SCADIS footprint model to compare turbulent flux footprints using different roughness lengths as in McGloin et al. (2014) is a good idea, however, we feel that such an in-depth analysis would distract from the broader goals of the paper. Although such a comparison would likely yield more data from the Long Point

site, our filtering based on an acceptable over-water wind direction of between 180-315 degrees likely ensured that only over-water fluxes were used in the analysis. The suggestion by the reviewer might also introduce some data that could be susceptible to land influence, thus compounding the effects of land contamination and over-lake model comparison, the latter being the main objective of this paper.

All the minor points have been corrected or revised as suggested. We appreciate the referee's careful review of the manuscript.

---

## Referee Comment (RC2) · Anonymous Referee #2 · 30 Apr 2018

General comments

The manuscript focuses on validation of modelled latent and sensible heat fluxes from the surface of the Great Lakes. Five algorithms from three parent models are tested against in situ data from Great Lakes Evaporation Network. It is in general a well written manuscript and a valuable contribution to the research field.

I would like to start with commenting the understandability of the abstract. I had to read the full manuscript before understanding the abstract. It is very technical, and I would suggest to refine the language to the extent that a non-expert can understand it. The introduction, however, is very well written and delivers the right message on why this

study is of importance. The method section is also easy to follow, although some parts gets very technical, especially page 10-11. I understand that technical details have to be provided, but I would suggest to refine the text to make it more understandable. Furthermore, this part is not my expertise, so I cannot comment on the correctness of the algorithms. The results section is easy to follow, but I think section 3.3 (Error dependence) fits better in Supplementary materials. The discussion section is mostly focused on the limitations of the study. Some parts that are summarized in Conclusion section could have been brought up already in discussions to balance up the discussion on the limitations.

Specific comments

[Page 7, Lines 25-26] Which version of COARE is used in the study – 2.6, 3.0 or 3.5?

[Page 14, Line 21] Remove the punctuation (.) after "into"

[Page 14, Line 24] Remove on of the "due to"

[Page 15, Line 6] Insert a "f" after "improvements o"

[Page 17, Line 2] Replace (:) with (.)

[Page 17, Line 4] Insert "section" before "3.3"

[Page 18, Lines 13-14] Remove "were filtered out" at the end of the sentence

[Page 33, Figure caption 1] Monitoring stations are referred to as lighthouse-based monitoring platforms when Fig. 1 is first brought up in the manuscript.
* * *

---

## Author Comment (AC3) · 15 May 2018

Paragraphs and comments that were written by the reviewer are indicated by an asterisk (*) at the beginning. Those without an asterisk are responses from the authors.

*General comments *The manuscript focuses on validation of modelled latent and sensible heat fluxes from the surface of the Great Lakes. Five algorithms from three parent models are tested against in situ data from Great Lakes Evaporation Network. It is in general a well written manuscript and a valuable contribution to the research field.

*I would like to start with commenting the understandability of the abstract. I had to read

the full manuscript before understanding the abstract. It is very technical, and I would suggest to refine the language to the extent that a non-expert can understand it. The introduction, however, is very well written and delivers the right message on why this study is of importance. The method section is also easy to follow, although some parts gets very technical, especially page 10-11. I understand that technical details have to be provided, but I would suggest to refine the text to make it more understandable. Furthermore, this part is not my expertise, so I cannot comment on the correctness of the algorithms. The results section is easy to follow, but I think section 3.3 (Error dependence) fits better in Supplementary materials. The discussion section is mostly focused on the limitations of the study. Some parts that are summarized in Conclusion section could have been brought up already in discussions to balance up the discussion on the limitations.

Thanks to the reviewer for the comments overall. Regarding the abstract, modifications have been made to attempt to make it clearer. One key thing based on your comments is that the term "original algorithms" is removed, since this language presupposes the alterations to the algorithms that we experimented with, but which are not mentioned until later.

Regarding the methods section, some small changes have been made to help with clarity. In a nutshell, eqs. (1) and (2) are the most generic form of bulk flux equations and are shared by all of the algorithms. Eqs. (3)-(7) fill in some more detail, but basically are also included in all of the algorithms. At this point, details of the formulation become more subjective in the choice of form of the equations (explaining the difference among the algorithms), which are followed by calibration of parameters. Eqs. (8)-(11) are used by some algorithms and not others, and more details are left to references to the original sources. Eq. (9) is used in COARE and is also essentially the difference between the "original" and "updated" versions of the other four algorithms. We did some work to untangle the disparate notations of the original sources to distill down to the common threads expressed in eqs. (1)-(7).

The authors consider the first paragraph of the Conclusions section as highly positive. Essentially, COARE stood out as giving the best simulation of fluxes, and when one aspect of COARE's formulation was inserted into the other algorithms, their simulations were greatly improved, while there was very low sensitivity to other differences among algorithms. Therefore, we feel that we have identified the main problem and solution. The second paragraph does not speak ill of this study, but does give a caution for the impending scenario when the flux algorithms are put back into the context of the models FVCOM, WRF, and LLTM. "On the other hand" is now removed from the beginning of that paragraph. The third and fourth paragraphs are attempts at explaining the reasons behind certain summertime discrepancies between modeled and observed fluxes. This is necessary, but a more positive spin is added at the beginning of the third paragraph by pointing out that the magnitude of cold-season fluxes is much larger and therefore influential on the annual energy budget. The final paragraph of the Conclusions section is specific to one of the stations, describing some of the problems with its site; we believe that we have done well in validating flux algorithms using the other stations.

*Specific comments *[Page 7, Lines 25-26] Which version of COARE is used in the study – 2.6, 3.0 or 3.5?

"It is equivalent to COARE 3.0" is inserted, because the original code was ported into FVCOM from COARE 2.6, but modifications have been made that are equivalent to COARE 3.0.

*[Page 14, Line 21] Remove the punctuation (.) after "into" *[Page 14, Line 24] Remove on of the "due to" *[Page 15, Line 6] Insert a "f" after "improvements o"

These previous three comments were already implemented in response to another set of comments.

*[Page 17, Line 2] Replace (:) with (.) *[Page 17, Line 4] Insert "section" before "3.3" *[Page 18, Lines 13-14] Remove "were filtered out" at the end of the sentence

These previous three comments were implemented in the manuscript exactly as suggested, and will appear in the final submission.

*[Page 33, Figure caption 1] Monitoring stations are referred to as lighthouse-based monitoring platforms when Fig. 1 is first brought up in the manuscript.

The caption now describes them as "offshore lighthouse-based monitoring stations".

The final submission will also incorporate some clarifications on the types of hardware at the stations, and clean up a few issues of notation.
* * *

---

## Author Response (AR1)

The comments by the reviewers are *italicized and blue-colored*. Our responses are in normal font and start with "Authors' response".
The manuscript with track changes is appended after the responses to the reviewers' comments. Some places edited to address the comments by the reviewers are yellow-highlighted for discoverability. The manuscript also incorporates typo fixes, and a few issues of notation. These changes are tracked as well.

**Responses to the comments by Dr. Tan**

*Dear Editor and Authors,*
*I'd like to leave some comments on this interesting study. In my opinion, its most obvious weakness is to use the satellite-derived water surface temperature to simulate sensible and latent heat fluxes. As water temperature is a sensitive variable for these simulations, the use of satellite-derived data for site-level applications may introduce large uncertainties, especially during the bad weather seasons when more data become unreliable.*

Authors' response: Indeed, using the satellite-derived water surface temperature (GLSEA) would be the most significant source of simulation errors. It is a composite analysis using cloud-free portions of satellite imageries and therefore the accuracy tends to be lower on cloudy days (discussed in Section 4). However, we confirmed that the GLSEA water surface temperature agreed reasonably with offshore buoy measurements during these periods. Even though offshore buoy sites are not collocated with the stations for eddy covariance measurements and buoy data are typically not available during December-March (mainly due to ice cover), the agreement between the GLSEA water surface temperature and buoy measurements provides us some confidence in the accuracy in the GLSEA water surface temperature. A recent study by Moukomla and Blanken (2016) tested an experimental method to derive water surface temperature from MODIS (or Moderate Resolution Imaging Spectroradiometer) for all-sky conditions. Such a product may also be tested in the future. This point was added at L12-15 in page 19 (yellow highlighted). However, as described in Section 4, in-situ measurements (e.g. thermistor) are desired for a more reliable simulation.

*Second, the use of daily water temperature and half-hourly meteorological variables simultaneously may also introduce additional errors, for example, during deep water mixing when energy balance is fast approached.*

Authors' response: The only hydrodynamic process that can cause rapid and sub-daily changes in the water surface temperature is upwelling, which are readily found in nearshore waters but not at the offshore locations of the eddy covariance stations. If the commenter is referring to deepwater mixing during fall overturn, that generally does not cause rapid temperature changes that would require sub-daily resolution. There- fore, daily water temperature would be sufficient for this study.

*Also, the authors mention that the flux algorithm (Z98L) of the WRF model is adapted from the CLM 4.5 where Z98L assumes that the roughness length scale of momentum, z0, is a constant 0.001 m and the roughness length scales of momentum, temperature and humidity are the same. Actually, in the lake model of CLM4.5 (CLM-CLISSS), z0 is updated dynamically and three roughness length scales are not equal (for frozen lakes, z0 is fixed). Please check with Subin et al. (2012) for accuracy.*

*Z. M.Subin, W. J.Riley and D.Mironov (2012), An improved lake model for climate simu- lations: Model structure, evaluation, and sensitivity analyses in CESM1, J. Adv. Model. Earth Syst., 4, M02001, doi:10.1029/2011MS000072.*

Authors' response: Thank you for pointing this out. The statement below was added at L21-23 in page 8 (yellow highlighted).

"…except that roughness length scales for temperature and humidity are constant for its WRF-lake application while they are updated dynamically in CLM 4.5."

**Responses to the comments by Anonymous Referee #1**

*This is generally a well written paper with a few minor grammatical errors. I have listed these below. My main issue is with the conclusions which read like a summary rather than conclusions. The main conclusion of this paper are that when corrected for $z_{\theta,q}$ the four models give similar results. I also wonder if the amount of measurement points that could be included would be helped by using the foot print analysis in McGloin et al. (2014, Water Resources Res., 50:494-513). This may allow more of the data at Long Point to be included.*

Authors' response: We renamed the title of Section 5 to "Summary and Conclusions", since this is more appropriate, as the referee pointed out. Also, we added to this section a sentence (yellow highlighted) "With the updated formula for $z_{0\theta,q}$, the four models (LS87, C89, J99, Z98L) simulated heat fluxes similar to COARE's"
The suggestion to use the SCADIS footprint model to compare turbulent flux footprints using different roughness lengths as in McGloin et al. (2014) is a good idea, however, we feel that such an in-depth analysis would distract from the broader goals of the paper. Although such a comparison would likely yield more data from the Long Point site, our filtering based on an acceptable over-water wind direction of between 180- 315 degrees likely ensured that only over-water fluxes were used in the analysis. The suggestion by the reviewer might also introduce some data that could be susceptible to land influence, thus compounding the effects of land contamination and over-lake model comparison, the latter being the main objective of this paper.

*Minor Corrections*

*1. Page 14, Line 11. Delete 'occurs'.*
Authors' response: Deleted.

*2. Page 14, lines 20-21. The sentence starting 'This is because...' does not make sense as written and has two full stops in it.*
Authors' response: Corrected (L7-9 in page 15, yellow highlighted). Thank you.

*3. Page 14, line 24. Delete the second 'due to'.*
Authors' response: Removed. Thank you.

*4. Page 15, line 6, Replace 'o' with 'of'.*
Authors' response: Added. Thank you.

*5. Page 15, lines25-26. I suggest rewriting after 'significant' with (28% compared to 50% for Stannard Rock, Spectacle Reef, and Long Point according to error reduction ratios (Table 3).*
Authors' response: Rewritten as "At White Shoal, however, the improvement was not as significant, 28 % compared to 57 % for Stannard Rock, 69 % for Spectacle Reef, and 50 % for Long Point according to error reduction ratios (Table 3). " (L15-18 in page 16, yellow highlighted).

*6. Page 15, line 28. Replace 'The' with 'These data'.*
Authors' response: Corrected. (L20 in page 16)

*7. Page 16, line 20. Space needed between 'sustained' and 'low'.*
Authors' response: Space added. (L12 in page 17)

*8. Page 17, line 17. 'more comprehensive.' in what? The authors need to elaborate how the models need to be improved and why.*
Authors' response: The sentence was updated to below (L13-16 in page 18):
"An ideal approach to improve model systems would have to be more comprehensive in terms of model variables of which a system is expected to provide best estimates. For example, in FVCOM, it may be a combination of improvements to a meteorological data set that drives the hydrodynamic model and to a bulk flux algorithm in it."

*9. Page 17, line 30. Insert 'from the present data set.' After 'possibility'.*
Authors' response: We decided to remove this phrase entirely as the discussed situation is unrealistic (L30 in page 18).

*10. Page 18, line 8. Replace '(i.e.' with 'and resulting in' and delete ')' after 'H'.*
Authors' response: Replaced. (L8 in page 19)

**Responses to the comments by Anonymous Referee #2**

*The manuscript focuses on validation of modelled latent and sensible heat fluxes from the surface of the Great Lakes. Five algorithms from three parent models are tested against in situ data from Great Lakes Evaporation Network. It is in general a well written manuscript and a valuable contribution to the research field.*
*I would like to start with commenting the understandability of the abstract. I had to read the full manuscript before understanding the abstract. It is very technical, and I would suggest to refine the language to the extent that a non-expert can understand it. The introduction, however, is very well written and delivers the right message on why this study is of importance. The method section is also easy to follow, although some parts gets very technical, especially page 10-11. I understand that technical details have to be provided, but I would suggest to refine the text to make it more understandable. Furthermore, this part is not my expertise, so I cannot comment on the correctness of the algorithms. The results section is easy to follow, but I think section 3.3 (Error dependence) fits better in Supplementary materials. The discussion section is mostly focused on the limitations of the study. Some parts that are summarized in Conclusion section could have been brought up already in discussions to balance up the discussion on the limitations.*

Authors' response: Thanks to the reviewer for the comments overall. Regarding the abstract, modifications have been made to attempt to make it clearer. One key thing based on the reviewer's comments is that the term "original algorithms" is removed, since this language presupposes the alterations to the algorithms that we experimented with, but which are not mentioned until later.
Regarding the methods section, some small changes have been made to help with clarity. In a nutshell, eqs. (1) and (2) are the most generic form of bulk flux equations and are shared by all of the algorithms. Eqs. (3)-(7) fill in some more detail, but basically are also included in all of the algorithms. At this point, details of the formulation become more subjective in the choice of form of the equations (explaining the difference among the algorithms), which are followed by calibration of parameters. Eqs. (8)-(11) are used by some algorithms and not others, and more details are left to references to the original sources. Eq. (9) is used in COARE and is also essentially the difference between the "original" and "updated" versions of the other four algorithms. We did some work to untangle the disparate notations of the original sources to distill down to the common threads expressed in eqs. (1)-(7).
 The authors consider the first paragraph of the Conclusions section as highly positive. Essentially, COARE stood out as giving the best simulation of fluxes, and when one aspect of COARE's formulation was inserted into the other algorithms, their simulations were greatly improved, while there was very low sensitivity to other differences among algorithms. Therefore, we feel that we have identified the main problem and solution. The second paragraph does not speak ill of this study, but does give a caution for the impending scenario when the flux algorithms are put back into the context of the models FVCOM, WRF, and LLTM. "On the other hand" is now removed from the beginning of that paragraph. The third and fourth paragraphs are attempts at explaining the reasons behind certain summertime discrepancies between modeled and observed fluxes. This is necessary, but a more positive spin

is added at the beginning of the third paragraph by pointing out that the magnitude of cold-season fluxes is much larger and therefore influential on the annual energy budget. The final paragraph of the Conclusions section is specific to one of the stations, describing some of the problems with its site; we believe that we have done well in validating flux algorithms using the other stations.

Finally, we think that section 3.3 should remain in the main manuscript, rather than supplemental materials. Some statements in the summary and conclusion section are supported by this section.

*Specific comments*
*[Page 7, Lines 25-26] Which version of COARE is used in the study – 2.6, 3.0 or 3.5?*
Authors' response: "It is equivalent to COARE 3.0" is inserted (L2-3 in page 8, yellow highlighted), because the original code was ported into FVCOM from COARE 2.6, but modifications have been made that are equivalent to COARE 3.0.

*[Page 14, Line 21] Remove the punctuation (.) after "into"*
Authors' response: Removed.

*[Page 14, Line 24] Remove on of the "due to"*
Authors' response: Removed.

*[Page 15, Line 6] Insert a "f" after "improvements o"*
Authors' response: These previous three comments were already implemented in response to another set of comments.

*[Page 17, Line 2] Replace (:) with (.)*
Authors' response: Replaced.

*[Page 17, Line 4] Insert "section" before "3.3"*
Authors' response: Inserted.

*[Page 18, Lines 13-14] Remove "were filtered out" at the end of the sentence*
Authors' response: Removed.

*[Page 33, Figure caption 1] Monitoring stations are referred to as lighthouse-based monitoring platforms when Fig. 1 is first brought up in the manuscript.*
Authors' response: The caption now describes them as "offshore lighthouse-based monitoring stations".

[revised manuscript text omitted]

momentum $z_0$[PB1]

$z_{0\theta,q}$

---

## Referee Report (RR1)

Review of Evaluating and improving modelled turbulent fluxes across the North American Great Lakes.

This is a well written paper and the authors have done a well at incorporating comments from my review and others. There are a couple of minor suggestions I have that may help to improve this and a couple of corrections that are needed.

I think in section 3.1 the authors need to give the long-term average temperatures when suggesting that the winter of 2012-2013 and 2013-2014 were unusually war4m and cold respectively. Similar with regard to the summer temperature in 2012. This would help to put these statements in perspective.

In section 3.2 line 21 it is suggested that the fluxes for Spectacle Reef are underestimated. I cannot see this in the figures, except when the $z_{0,\theta,q}$ is updated. This needs to be rewritten so that the statement matches the figure.

The last sentence of the last paragraph of section 3.2 suggests that for White Shoal there was under- and overestimation, but this is very hard to see in the figures. I suggest you either use an insert or put a larger figure in the supplementary material.

I wonder if counter-gradient flux (Deardoff, 1966, Am. Mereological Soc., 23: 503-506) is an explanation for the discrepancy discussed on page 18 first paragraph.

I found it difficult to see the differences between the original and updated data in figures 8-11. I wonder if using a x for one and an open circle for the other would be better.

Minor Corrections.
1. Page 11, line 5. Beljaars 1994 need to be a or b citation.
2. Page 17, line 16. Beljaars 1994 need to be a or b citation.
3. Page 18, line 12. Replace 'are' with 'is' and insert 'to be' after 'likely'.
4. Page 20, line 20. Insert 'Lake' before 'Communities'.
5. Table 1, Z98L, COARE. Citation of Fairall et al. (1996) needs to be a or b or both.
6. Page 25. Holtslag et al. (1991) is not in reference list but appears multiple times in text and tables.
7. Page 28. Reference Smith (1989) is not cited.
8. Page 29. Reference Zeng (2002) is not cited.

---

## Author Response (AR2)

The comments by the reviewer are *italicized and blue-colored*. Our responses are in normal font and start with "Authors' response". The revised manuscript with tracked changes follows after these responses to the reviewer's comments.

While revising the manuscript, the authors found that the sensor height for Stannard Rock should be 39 m instead of 32.5m that was used in the previous manuscript. Also, there was a minor error in bias calculation for Stannard Rock. The simulations and analyses for Stannard Rock were re-run and the statistics in Table 2 was updated (Only a few W/m2 difference in RMSEs). The timeseries of the heat fluxes look nearly identical. No change in the conclusions). These changes, as well as the changes to address the reviewer's comments, are tracked.

*Review of Evaluating and improving modelled turbulent fluxes across the North American Great Lakes.*

This is a well written paper and the authors have done a well at incorporating comments from my review and others. There are a couple of minor suggestions I have that may help to improve this and a couple of corrections that are needed.

Authors' response: Thank you again for your helpful review and suggestions. The responses to the comments are listed below.

I think in section 3.1 the authors need to give the long-term average temperatures when suggesting that the winter of 2012-2013 and 2013-2014 were unusually war4m and cold respectively. Similar with regard to the summer temperature in 2012. This would help to put these statements in perspective.

Authors' response: Thank you for the comment. This is indeed helpful for readers. We included the mean surface air temperatures for the winters of 2012-2013 and 2013-2014, and for the spring of 2012, as well as the long-term (1948-2014) average temperatures for the same months (section 3.1, yellow highlighted). The description for the data source is also added to the end of section 2.1.2 (yellow highlighted).

In section 3.2 line 21 it is suggested that the fluxes for Spectacle Reef are underestimated. I cannot see this in the figures, except when the  $z0, \theta, q$  is updated. This needs to be rewritten so that the statement matches the figure.

Authors' response: The sentence "This is also represented in the 10-day running mean time series as well (Fig. 6ab)." was added in section 3.1 (yellow highlighted).

The last sentence of the last paragraph of section 3.2 suggests that for White Shoal there was under- and overestimation, but this is very hard to see in the figures. I suggest you either use an insert or put a larger figure in the supplementary material.

Authors' response: The sentence only refers to the underestimation at White Shoal (not overestimation). We suspect the reviewer misread it. To improve the readability, the references to Figs. 4,5,8, and 9 are added.

*I wonder if counter-gradient flux (Deardoff, 1966, Am. Mereological Soc., 23: 503-506) is an explanation for the discrepancy discussed on page 18 first paragraph.*

Authors' response: Thank you for the comment. We think the counter-gradient flux would unlikely be a possible explanation here, especially as nonlocal motion of air parcels should not persist for such a long period (> 1 month) but has shorter timescales (

**Minor Corrections.**

1. Page 11, line 5. Beljaars 1994 need to be a or b citation.

2. Page 17, line 16. Beljaars 1994 need to be a or b citation.

Authors' response: There was only one Beljaars 1994 (i.e. no a or b is needed) but References list two Beljaars 1994. This was fixed. Thank you.

**3. Page 18, line 12. Replace 'are' with 'is' and insert 'to be' after 'likely'.**

Authors' response: This sentence was revised as below (yellow highlighted).

"However, evidence to confirm these possibilities is not sufficient at this time"

**4. Page 20, line 20. Insert 'Lake' before 'Communities'.**

Authors' response: Thank you. This sentence was rewritten as below (yellow highlighted). "Communities within and surrounding the Great Lakes basin are increasingly dependent on numerical geophysical models for these types of societal applications."

5. Table 1, Z98L, COARE. Citation of Fairall et al. (1996) needs to be a or b or both. Authors' response: Both of "ab" are added. Thank you.

**6. Page 25. Holtslag et al. (1991) is not in reference list but appears multiple times in text and tables.**

Authors' response: There are "Beljaars and Holtslag (1991)" and "Holtslag (1990)" in text and they are indeed listed in reference, but there is no "Holtslag (1991)" both in text and reference. Perhaps the reviewer took "Beljaars and Holtslag (1991)" as "Holtslag (1991)". No change is made for this point.

**7. Page 28. Reference Smith (1989) is not cited.**

Authors' response: Smith (1989) was removed from the reference list, thank you.

**8. Page 29. Reference Zeng (2002) is not cited.**

Authors' response: Zeng (2002) was removed from the reference list, thank you.

**Evaluating and improving modeled turbulent heat fluxes across the North American**

**Great Lakes**

Umarporn Charusombat1, Ayumi Fujisaki-Manome2,3, Andrew D. Gronewold1, Brent M. Lofgren1, Eric J. Anderson1, Peter D. Blanken4, Christopher Spence5, John D. Lenters6, 5 Chuliang Xiao2, Lindsay E. Fitzpatrick2, and Gregory Cutrell7

[revised manuscript text omitted]

| Deleted:
Deleted:                                                                                                                                                                                                                                                                                                                                                                                                                                                                                                                                                                                                                                                                                                                                                                                                                                                                                                                                                                                                                                                                                                                                                                                                                                                                                                                                                                                                                                                                                                                                                                                                                                                                                                                                                                                                                                                                                                                                                                                                                                                                            |            |  |
|-----------------------------------------------------------------------------------------------------------------------------------------------------------------------------------------------------------------------------------------------------------------------------------------------------------------------------------------------------------------------------------------------------------------------------------------------------------------------------------------------------------------------------------------------------------------------------------------------------------------------------------------------------------------------------------------------------------------------------------------------------------------------------------------------------------------------------------------------------------------------------------------------------------------------------------------------------------------------------------------------------------------------------------------------------------------------------------------------------------------------------------------------------------------------------------------------------------------------------------------------------------------------------------------------------------------------------------------------------------------------------------------------------------------------------------------------------------------------------------------------------------------------------------------------------------------------------------------------------------------------------------------------------------------------------------------------------------------------------------------------------------------------------------------------------------------------------------------------------------------------------------------------------------------------------------------------------------------------------------------------------------------------------------------------------------------------------------------------------------------------------------------|------------|--|
| Deleted: -
| Deleted:
Deleted:                                                                                                                                                                                                                                                                                                                                                                                                                                                                                                                                                                                                                                                                                                                                                                                                                                                                                                                                                                                                                                                                                                                                                                                                                                                                                                                                                                                                                                                                                                                                                                                                                                                                                                                                                                                                                                                                                                                                                                                                                                                                                                      | Deleted: - |  |
| Deleted: N Deleted:                                                                                                                                                                                                                                                                                                                                                                                                                                                                                                                                                                                                                                                                                                                                                                                                                                                                                                                                                                                                                                                                                                                                                                                                                                                                                                                                                                                                                                                                                                                                                                                                                                                                                                                                                                                                                                                                                                                                                                                                                                                                                                                     | Deleted:   |  |
| Deleted:                                                                                                                                                                                                                                                                                                                                                                                                                                                                                                                                                                                                                                                                                                                                                                                                                                                                                                                                                                                                                                                                                                                                                                                                                                                                                                                                                                                                                                                                                                                                                                                                                                                                                                                                                                                                                                                                                                                                                                                                                                                                                                                                | Deleted: N |  |
| Deleted.                                                                                                                                                                                                                                                                                                                                                                                                                                                                                                                                                                                                                                                                                                                                                                                                                                                                                                                                                                                                                                                                                                                                                                                                                                                                                                                                                                                                                                                                                                                                                                                                                                                                                                                                                                                                                                                                                                                                                                                                                                                                                                                                | Deleted:   |  |
| Deletea:                                                                                                                                                                                                                                                                                                                                                                                                                                                                                                                                                                                                                                                                                                                                                                                                                                                                                                                                                                                                                                                                                                                                                                                                                                                                                                                                                                                                                                                                                                                                                                                                                                                                                                                                                                                                                                                                                                                                                                                                                                                                                                                                | Deleted:   |  |

| - | Deleted: differed |
|---|-------------------|
|   | Deleted:          |
|   | Deleted:          |

[... [1]]

closest pixel to a GLEN station was greater than zero, we did not use any data for our comparison (i.e. the observed heat fluxes, water surface temperature, and meteorological data). This was because the study focused on evaluating the turbulent heat fluxes over water during ice-free periods.

- 5 Infrared thermometers (IRTs, Apogee IRR-T) were also installed on the observation platforms to measure water surface temperature. However, test simulations showed that the flux values simulated using the water surface temperature from the IRTs were generally less reliable than when using the GLSEA data. Blanken et al. (2011) found that about 30% of the IRT-measured lake surface temperature observations were unreliable due to condensation, frost, and
- 10 interference from other surfaces (e.g., the lighthouse or sky). It is likely that this issue affected the accuracy of IRT-measured water surface temperature during the period of our study. Therefore, we did not use the IRT-based measurements of water surface temperature as input to the simulations.

Monthly surface air temperature over the Great Lakes is used in the text as a measure of anomalously warm and cold seasons. These data are taken from the Great Lakes Monthly Hydrologic Data (https://www.glerl.noaa.gov/ahps/mnth-hydro.html).

**2.2. Flux algorithms**

20

15

We evaluated five different flux algorithms that are incorporated into the three hydrodynamic/atmospheric/hydrologic models that are frequently used for Great Lakes operational and research applications (Fig. 2).

In an early stage of its development, FVCOM required prescribed heat fluxes as forcing variables, rather than being calculated (Chen et al., 2006a,b). In a subsequent version of FVCOM (Version 2.7), turbulent fluxes were calculated using the Coupled Ocean Atmosphere Response

25 Experiment (COARE) Met Flux Algorithm version 2.6 (Fairall et al., 1996a,b), which was first adopted in the official FVCOM by (Chen et al., 2006a). The COARE Met Flux Algorithm is one of the most frequently used algorithms in the air-sea interaction community. It was subsequently modified and validated at higher winds in the version known as COARE 3.0 (Fairall et al., 2003) and the latest version COARE 3.5 (Edson et al., 2013), which includes wave influences on the Charnock parameter (Charnock, 1955). FVCOM mostly incorporated these updates as the model was upgraded, including provision for freshwater implementation, except that the latest version

7

| Deleted: were e  | valuated |  |
|------------------|----------|--|
|                  |          |  |
|                  |          |  |
|                  |          |  |
| Deleted: ,       |          |  |
| Deleted: not     |          |  |
| Deleted: , varia | oles     |  |
| Deleted: b       |          |  |
| Deleted: 2006    |          |  |
| Deleted: a       |          |  |
| Deleted.         |          |  |

[revised manuscript text omitted]
 -2 ]           COARE         J99         LS87         Z98L         C89         red |                              |                              |                                        | Error                        | Mean flux [W    | Mean              | Mean bias [%]       |                            |
|----------------------------------|--------------------------------------------------------------------------------------------------------|------------------------------|------------------------------|----------------------------------------|------------------------------|-----------------|-------------------|---------------------|----------------------------|
|                                  | COARE                                                                                                  | J99                          | LS87                         | Z98L                                   | C89                          | reduction ratio | m -2 ] | Normalized          |                            |
|                                  |                                                                                                        |                              |                              |                                        |                              | [%]             |                   | RMSE                |                            |
| Stannard Rock                    | 26.3                                                                                                   | 33. 7 (3 1.0 ) | 28 .3 (3 7.2 ) | 2 8 .1 (7 6 . 7 ) | 2 8.2 (3 6.8 ) | 3 5 .0   | 56.9              | 0.53 (0.84)         | 1.8 (3 1 .3) |
| White Shoal                      | 25.2                                                                                                   | 36. (25.3)                   | 28.3 (25.4)                  | 27.8 (68.0)                            | 27.6 (25.8)                  | 17.0            | 61.1              | 0.49 (0.59)         | 1.4 (24.0)                 |
| Spectacle Reef                   | 70.4                                                                                                   | 83.8 (66.8)                  | 68.5 (61.9)                  | 67.4 (72.6)                            | 71.3 (62.5)                  | -10.3           | 116.1             | 0.63 (0.57)         | -27.8 (-3.2)               |
| Long Point                       | 42.9                                                                                                   | 40.1 (42.1)                  | 47.9 (46.5)                  | 49.1 (104.3)                           | 45.8(47.8)                   | 24.1            | 50.7              | 0.90 (1.19)         | 27.4 (49.6)                |
| Mean RMSE
[Wm -2 ] | 41. 2                                                                                           | 48. 5 (41. 3 ) | 43. 2 (4 2.8 ) | 43. 1 (8 0.4 )           | 43. 2 (43. 2 ) | 1 4.3    | 81.5              | 0.5 5 (0.64) | -                          |

5.5 (17.0)

-

12.4 (91.3)

**Table 3** Same as Table 2, but for sensible heat flux *H*.

-23.5 (2.5)

11.7 (16.2)

Mean bias [%]

-2.4

|                | RMSE [Wm -2 ] |                              |                              |                               |                              | Error reduction | Mean flux [W      | Normalized                   | Mean bias [%]               |
|----------------|--------------------------|------------------------------|------------------------------|-------------------------------|------------------------------|-----------------|-------------------|------------------------------|-----------------------------|
|                | COARE                    | J99                          | LS87                         | Z98L                          | C89                          | ratio [%]       | m -2 ] | RMSE                         |                             |
| Stannard Rock  | 2 5.1             | 2 7.2 ( 47.8 ) | 2 4.5 (8 1.0 ) | 2 4 .5 (7 3 .4) | 2 2.0 ( 29.7 ) | 5 7.6    | 39.1              | 0.6 3 (1. 48 ) | -8.9 ( 36.3 ) |
| White Shoal    | 32.3                     | 31.4 (37.9)                  | 31.8 (50.8)                  | 31.9 (52.8)                   | 31.0 (32.9)                  | 27.7            | 40.7              | 0.78 (1.07)                  | -24.9 (7.8)                 |
| Spectacle Reef | 11.4                     | 13.2 (27.2)                  | 13.9 (60.4)                  | 11.9 (65.3)                   | 13.3 (13.8)                  | 68.6            | 46.1              | 0.28 (0.90)                  | 6.3 (44.8)                  |
| Long Point     | 27.2                     | 26.7 (45.5)                  | 28.5 (65.6)                  | 27.6 (63.2)                   | 21.5 (32.9)                  | 49.7            | 11.7              | 2.2 (4.4)                    | 18.5 (31.4)                 |
| Mean RMSE      | 24. 0             | 2 4.7 ( 39.6 ) | 2 4.7 (6 4.5 ) | 24. 0 (6 3 .7)  | 22. 0 (2 7.4 ) | 51. 2    | 38.0              | 0.6 3 (1. 28 ) | -                           |
| Mean bias [%]  | -5.6              | -3.3 (25.8)           | 8.4 (61. 4 )   | -2.5 (58.2 )           | -8.3 (4.9 )           | -               | -                 | -                            | -2.3 ( 30.1 ) |

**Captions of Figures**

Figure 1. Map of the Laurentian Great Lakes including the locations of offshore lighthouse-based monitoring stations used in this study. Adapted from Lenters et al. (2013). Instrument heights above the mean water level are 39, m at Stannard Rock, 29.5 m at Long Point, 30.0 m at Spectacle Reef, and 42.8 m at White Shoal.

Figure 2. Schematic diagram showing the relationship between the parent model systems (FVCOM, WRF-Lake, and LLTM) and the flux algorithms used in the parent model systems. Detail description of each flux algorithm is listed in Table 1.

Figure 3. 10-day running mean time series of meteorological variables at the four stations. Air temperature and relative humidity were measured with Vaisala HMP45C thermohygrometers and wind speed were measured with the CSAT-3 (See section 2.1.1 or Figure 1 for the sensor heights). Water surface temperature is taken from GLSEA. Data at pixels closest to the stations are used. The data gaps in water surface temperature from January to April denote periods during which the site was affected by lake ice cover. Measurements at Long Point and White Shoal started in May and June of 2012. There is also a long data gap between February 2012 and June 2013 at Spectacle Reef.

Figure 4. 10-day running mean time series of latent ( $\lambda E$ ) and sensible (*H*) heat fluxes at Stannard Rock. Black lines denote observed  $\lambda E$  and *H* and the same for (a), (b) and (c), (d), respectively. The  $\lambda E$  and *H* simulations employ the original  $z_{\partial \theta,q}$  formula in (a), (c) and with the updated  $z_{\partial \theta,q}$  formula in (b) and (d). The COARE simulation results are unchanged from (a) to (b) or from (c) to (d).

Figure 5. The same as Figure 4, but at White Shoal.

Figure 7. The same as Figure 4, but at Long Point.

Figure 6. The same as Figure 4, but at Spectacle Reef.

Figure 8. Scatter plots of latent heat flux ( $\lambda E$ ) comparing the observed (*x*-axis) and the simulated (*y*-axis) daily mean fluxes. Each row shows comparisons with a specific

33

algorithm at the four stations, while each column shows comparisons with the five algorithms at a specific station. Grey and blue dots indicate the results with the original and updated  $z_{\partial \theta,q}$  formulae, respectively.

Figure 9. The same as Figure 8, but for sensible heat flux (*H*).

Figure 10. Errors in daily mean latent heat flux (y-axis) versus specific humidity difference between the water surface and air at the sensor height  $q_{\cdot}-q_{\epsilon}$  [kg kg\*], transfer coefficient  $C_{\epsilon}$ [-], wind speed U [m s4], and stability factor z/L (x-axis) for the five algorithms at Stannard Rock. Grey and blue dots indicate the results using the original and updated  $z_{eq_{\epsilon}}$  formulae, respectively.

Figure 11. Errors in daily mean sensible heat flux (y-axis) versus potential temperature difference between the water surface and air at the sensor height  $\theta_w$ - $\theta_a$  [·C], transfer coefficient  $C_a$  [-], wind speed U [m s1], and stability factor z/L (x-axis) for the five algorithms at Stannard Rock. Grey and blue dots indicate the results with the original and updated  $z_{aa}$  formulae, respectively.